# ForensicHub: A Unified Benchmark & Codebase for All-Domain Fake Image Detection and Localization

**Bo Du**[1†], **Xuekang Zhu**[1,2†], **Xiaochen Ma**[3†], **Chenfan Qu**[4, 2†], **Kaiwen Feng**[1†],
**Zhe Yang**[1], **Chi-Man Pun**[5], **Jian Liu**[2‡], **Ji-Zhe Zhou**[1‡]

[1]Sichuan University, [2]Ant Group, [3]HKUST
[4]South China University of Technology, [5]University of Macao

## Abstract

The field of Fake Image Detection and Localization (*FIDL*) is highly fragmented, encompassing four domains: deepfake detection (Deepfake), image manipulation detection and localization (IMDL), artificial intelligence-generated image detection (AIGC), and document image manipulation localization (Doc). Although individual benchmarks exist in some domains, a unified benchmark for all domains in FIDL remains blank. The absence of a unified benchmark results in significant domain silos, where each domain independently constructs its datasets, models, and evaluation protocols without interoperability, preventing cross-domain comparisons and hindering the development of the entire FIDL field. To close the domain silo barrier, we propose ForensicHub, the first unified benchmark & codebase for all-domain fake image detection and localization. Considering drastic variations on dataset, model, and evaluation configurations across all domains, as well as the scarcity of open-sourced baseline models and the lack of individual benchmarks in some domains, ForensicHub: i) proposes a modular and configuration-driven architecture that decomposes forensic pipelines into interchangeable components across datasets, transforms, models, and evaluators, allowing flexible composition across all domains; ii) fully implements 10 baseline models (3 of which are reproduced from scratch), 6 backbones, 2 new benchmarks for AIGC and Doc, and integrates 2 existing benchmarks of DeepfakeBench and IMDLBenCo through an adapter-based design; iii) establishes an image forensic fusion protocol evaluation mechanism that supports unified training and testing of diverse forensic models across tasks; iv) conducts indepth analysis based on the ForensicHub, offering 8 key actionable insights into FIDL model architecture, dataset characteristics, and evaluation standards. Specifically, ForensicHub includes 4 forensic tasks, 23 datasets, 42 baseline models, 6 backbones, 11 GPU-accelerated pixel- and image-level evaluation metrics, and realizes 16 kinds of cross-domain evaluations. ForensicHub represents a significant leap forward in breaking the domain silos in the FIDL field and inspiring future breakthroughs. Code is available at: `https://github.com/scu-zjz/ForensicHub`.

## 1 Introduction

"*The whole is more than the sum of its parts*" - ***Aristotle***

Fake images have become increasingly prevalent, driven by the rapid advancement of various digital image editing techniques in recent years. This highlights the importance of Fake Image Detection and

---

[†]Equal contribution.
[‡]Corresponding authors : Jian Liu (rex.lj@antgroup.com) and Ji-Zhe Zhou (jzzhou@scu.edu.cn)

39th Conference on Neural Information Processing Systems (NeurIPS 2025) Track on Datasets and Benchmarks.

Table 1: Summary of representative methods from four forensic domains, detailing model design, backbone, artifact strategy, output format, and core contributions.

| Task | Model | Backbone | Artifact Strategy | Output Type | Contribution |
|---|---|---|---|---|---|
| Deepfake | Capsule-Net [42](ICASSP19) | VGG [55] | Dynamic Routing | Label | Proposes a capsule network with dynamic routing and a VGG19 backbone. |
| | RECCE [4](CVPR22) | Xception [8] | Reconstruction | Label | Proposes a graph-based framework leveraging reconstruction differences |
| | SPSL [32](CVPR21) | Xception [8] | Phase Spectrum | Label | Proposes phase-spectrum fusion with Xception for face forgery detection. |
| | UCF [75](ICCV23) | Xception [8] | Multi-task Disentanglement | Label | Proposes multi-task disentanglement with Xception for deepfake generalization. |
| | SBI [54](CVPR22) | EfficientNet [61] | Frequency,Blending Boundaries | Label | Proposes self-blended images to improve deepfake detection generalization. |
| IMDL | MVSS-Net [68](ICCV21) | Resnet [21] | BayarConv,Sobel | Label,Mask | Exploit noise and boundary artifacts via multi-view learning for manipulation detection. |
| | CAT-Net [24](IJCV22) | HRNet [62] | DCT | Mask | Fuse RGB and DCT streams to learn compression artifacts for splice localization. |
| | PSCC-Net [34](TCSVT22) | HRNet [62] | Multi-Resolution Conv | Label,Mask | Progressively refine masks with spatio-channel correlations for high-reso localization. |
| | Trufor [18](CVPR23) | Seformer [70] | High-Reso,Multi-scale,Edge | Label,Mask | Fuse RGB and learned noise fingerprints to detect manipulations as anomalies. |
| | IML-ViT [40](Arxiv) | ViT [14] | None | Mask | Use ViT with high-reso, multi-scale edge-aware design for manipulation localization. |
| | Mesorch [83](AAAI25) | Conv. [36],Segfor. [70] | DCT | Mask | Fuse micro and macro cues for mesoscopic image manipulation localization. |
| AIGC | Dire [65](ICCV23) | Resnet [21] | Diffusion Reconstruction | Label | Use reconstruction error of diffusion for diffusion-generated images detection. |
| | DualNet [69](APSIPA23) | CNN | SRM,Low Frequency | Label | Fuse SRM residual and low-frequency content streams for AIGC detection. |
| | HiFiNet [19](CVPR23) | HRNet [62] | Multi-branch Feature Extractor | Label,Mask | Learn hierarchical fine-grained representations of forgery attributes. |
| | Synthbuster [2](OJSP23) | None | Fourier Transform | Label | Leverage spectral artifacts in the frequency domain for diffusion detection. |
| | UnivFD [45](CVPR23) | CLIP-ViT [50] | None | Label | Use pretrained vision-language model features for unified detection. |
| Document | CAFTB [57](TOMM24) | Resnet [21] | SRM | Mask | Proposes CAFTB-Net with dual-branch and cross-attention. |
| | TIFDM [13](TCE24) | Resnet [21] | None | Mask | Proposes a robust network with multiscale attention. |
| | DTD [48](CVPR23) | Conv. [36], Swin.[35] | Frequency | Mask | Proposes DTD with frequency head and multi-view decoder |
| | FFDN [6](ECCV24) | ConvNext [36] | Wavelet, Frequency | Mask | Proposes FFDN combining visual enhancement and frequency decomposition |

Localization (FIDL), which aims to distinguish partially tampered and fully generated images from real ones. In FIDL, the term *Detection* refers to classification at the image level, while *Localization* targets a finer-grained segmentation of manipulated pixels at the pixel level.

However, the research efforts of FIDL have gradually split into four relatively independent research domains over time. 1) **Deepfake detection (*Deepfake*)** [42, 29, 27, 9, 43, 4, 75, 20, 47, 32, 38, 59, 84, 54, 72, 76]: *detects* human-centric manipulations such as face swapping, expression editing, or feature replacement. 2) **Image manipulation detection/localization (*IMDL*)** [31, 5, 18, 34, 81, 40, 78, 83, 58]: *detects and localizes* the tampering in natural images. 3) **AI-Generated Image Detection (*AIGC*)** [45, 19, 69, 65, 79]: *detects* the images fully generated by deep generative models such as Stable Diffusion [51]. 4) **Document Image Manipulation Localization (*Document*)** [53, 48, 6, 13, 57, 77, 28]: *localizes* the forgery of various forms of document images, including receipts, certificates, and identification materials, with a particular focus on detecting modifications to the printed text.

Although these domains have become isolated due to differences in application scenarios, manipulation types, and detection methods, there are still overlaps and similarities among them. As vision tasks, these four domains almost universally adopt SoTA detection or segmentation models as pre-trained backbones. Further, since the creators of fake images typically aim to preserve semantically plausible and realistic content, all four domains have placed considerable emphasis on designing low-level visual feature extractors to capture subtle, non-semantic discrepancies for reliable detection. Some research methodologies, such as contrastive learning, are commonly employed across these areas to mine discriminative features.

We summarize SoTAs in four domains of the backbone, artifacts strategy, output type, and contribution in Table 1. The differences cause the four FIDL domains to become fragmented, but the similarities call for a unified perspective to understand them cohesively.

Although individual benchmarks exist in some domains, such as *DeepfakeBench* [76] for Deepfake and *IMDLBenCo* [41] for IMDL, a unified benchmark for all domains in FIDL remains blank. The absence of such a unified benchmark results in significant **domain silos**, where each domain independently constructs its datasets, models, and evaluation protocols without interoperability. Domain silos lead to redundant and uneven research across existing FIDL fields, and difficulty in establishing a general and unified FIDL approach, severely hindering the development of the entire FIDL field.

Besides, in real-world scenarios, it is often impossible to predetermine the type of manipulation (deepfake, imdl, aigc and document) present in an image, making unified detection particularly important for users.

Therefore, establishing a unified benchmark for all domains is critically significant. However, such a benchmark faces the following challenges. Firstly, the drastic variations in datasets, models, and evaluation configurations across all domains require the benchmark to be sufficiently extendable and flexible in its design to support all domains. Secondly, compatibility with existing benchmarks is needed to reduce redundant research, while also addressing the scarcity of open-sourced baseline models and the absence of individual benchmarks in certain domains.

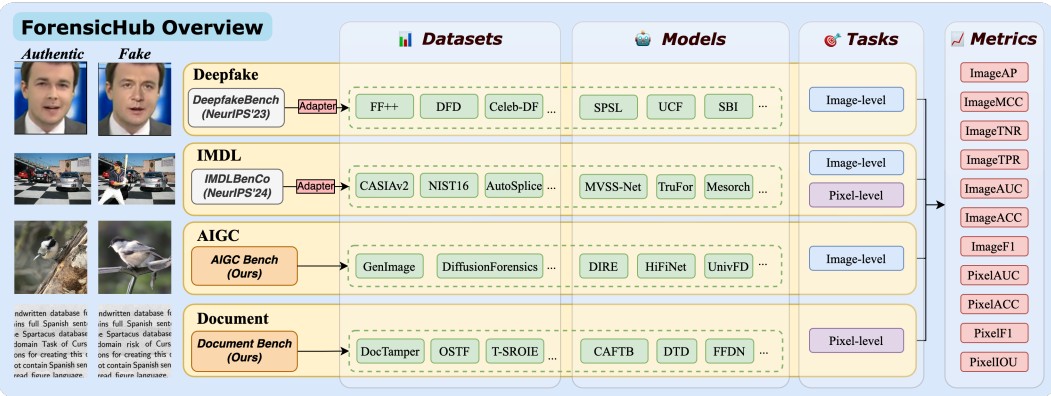

Figure 1: Overview of our ForensicHub. It is compatible with DeepfakeBench and IMDLBenCo via adapters, and introduces new AIGC and Document benchmarks. ForensicHub allows datasets and models from any domain to be freely combined into custom pipelines.

To this end, we propose **_ForensicHub_**, which: 1) proposes a modular and configuration-driven architecture that decomposes forensic pipelines into interchangeable components across datasets, transforms, models, and evaluators, allowing flexible composition across all domains; 2) fully implements 10 baseline models (3 of which are reproduced from scratch), 6 backbones, 2 new benchmarks for AIGC and Doc, and integrates 2 existing benchmarks of DeepfakeBench and IMDLBenCo through an adapter-based design.

With the above efforts, ForensicHub serves as the first unified benchmark and codebase for all-domain fake image detection and localization. Building on ForensicHub, we establish an image forensic fusion protocol *(IFF-Protocol)* evaluation mechanism that supports unified training and testing of diverse forensic models across tasks. We conduct deep analysis on 8 issues that are of particular interest to researchers but have not yet been thoroughly investigated, offering new insights into FIDL model architecture, dataset characteristics, and evaluation standards. The introduction of ForensicHub bridges all domains within FIDL, breaks down domain silos, and inspires future breakthroughs.

## 2 Related Works

Fake image detection and localization encompass four sub-tasks: 1) Deepfake Detection, 2) Image Manipulation Detection and Localization, 3) AI-Generated Image Detection, and 4) Document Image Manipulation Localization. The characteristics of each task are summarized in Appendix C. Despite rapid progress, a unified benchmark is lacking—each task uses isolated pipelines, limiting cross-task comparison.

Despite the rapid development of these tasks, there is a lack of a unified benchmark, with some task having its isolated benchmark, creating barriers between them.

DeepfakeBench [76] is a Deepfake detection benchmark specifically designed to address the lack of uniformity in data processing pipelines, leading to inconsistent data inputs for detection models. IMDLBenCo [41] is a benchmark and codebase for IMDL, aiming to compare IMDL models through a unified training and evaluation protocol. AIGCDetectBenchmark [80] is a repository for experiments on 9 AI-generated image detection methods.

These benchmarks provide models, datasets, and evaluation metrics within their respective tasks, but their underlying designs lack cross-task considerations, making them difficult to integrate across different detection scenarios. For example, DeepfakeBench is tightly coupled with Deepfake-specific data preprocessing steps, such as facial landmarks, while IMDLBenCo requires both datasets and models to output pixel-level masks. AIGCDetectBenchmark does not handle multi-GPU metric computation effectively. Additionally, none of them include a comprehensive set of image-level and pixel-level metrics. These limitations call for a new, unified, and flexible cross-task benchmark.

# 3 ForensicHub

In this section, we present our ForensicHub, which is a unified benchmark for all-domain fake image detection and localization designed for flexibility and extensibility, as illustrated in Figure 1.

**Modular Architecture.** To accommodate different forensic tasks, ForensicHub is designed as a modular architecture consisting of four main components: *Datasets, Transforms, Models, and Evaluators*. 1) *Datasets* handle the data loading process and are required to return fields that conform to the ForensicHub specification. 2) *Transforms* handle the data pre-processing and augmentation for different tasks. 3) *Models*, through alignment with *Datasets* and unified output, allow for the inclusion of various state-of-the-art image forensic models. 4) *Evaluators* cover commonly used image- and pixel-level metrics for different tasks, and are implemented with GPU acceleration to improve evaluation efficiency during training and testing.

**Configurable workflow.** ForensicHub provides a codeless approach for users to build training or testing workflows directly through the configuration of YAML files. Based on the modular architecture, users can select different evaluators to train and test ***any model*** on ***any dataset***. ForensicHub also provides a code generator for customized purposes, allowing users to integrate with the benchmark with minimal coding effort.

**Construction of ForensicHub.** To enable broad interoperability and reduce duplication of effort, ForensicHub adopts an *adapter-based design* [15] that ensures seamless integration with *DeepfakeBench* [76] and *IMDLBenCo* [41], two widely used benchmarks. This mechanism allows users to reuse existing models and datasets without major modification, while also supporting the definition of new models and benchmarks within ForensicHub under the unified protocol. This unified infrastructure simplifies cross-task benchmarking, supports reproducibility, and enables consistent evaluation across domains.

Specifically, ForensicHub supports all 10 models from IMDLBenCo for multi-domain and cross-domain evaluation. From DeepfakeBench, 27 out of 34 image-level detectors are compatible, including 5 general-purpose backbones and 9 domain-specific models that are not applicable to cross-task evaluation. The remaining 13 models support training or inference across different forensic domains. Therefore, 22 models of DeepfakeBench are included. ForensicHub fully implements 5 baseline models for AIGC and 5 baseline models for Document, with details in Sec. 4. In addition, ForensicHub includes 6 commonly used backbones. In total, ForensicHub covers 4 tasks, 23 datasets, 42 models, 6 backbones, and implements 11 commonly used image- and pixel-level metrics.

**Datasets** used in this paper are: FaceForensics++ [52], Celeb-DF [30], DFD [10], FaceShifter [25] and UADFV [29] for Deepfake; CASIA [12], COVERAGE [66], Columbia [22], IMD2020 [44], NIST16 [17], CocoGlide [18], and Autosplice [23] for IMDL; DiffusionForensics [65], GenImage [82] for AIGC; Doctamper [48], T-SROIE [64], OSTF [49], TPIC-13 [63], RTM [37] for Doc. A brief summary of each dataset is provided in Table 2, with more details in Appendix D.1.

**Models** used in this paper are: Capsule-Net [42], RECCE [4], SPSL [32], UCF [75], and SBI [54] for Deepfake detection; MVSS-Net [5], CAT-Net [24], PSCC-Net [34], Trufor [18], IML-ViT [40], and Mesorch [83] for image manipulation and localization; Dire [65], DualNet [69], HiFiNet [19], Synthbuster [2], and UnivFD [45] for AIGC detection; DTD [48], FFDN [6], CAFTB [57], TIFDM [13] for document detection. These methods are from official repositories and our reimplementations. In addition, ForensicHub also selects 6 commonly used backbones in visual tasks, which are: Resnet [21], Xception [8], EfficientNet [61], Segformer [70], Swin Transformer [35], and ConvNext [36]. Details about models can be found in Appendix D.2.

**Metrics** used in this paper are: AP, MCC, TNR, TPR, AUC, ACC, F1, and IOU, with pixel- and image-level implementations shown in Fig. 1. Details of each metric can be found in Appendix D.3. In the evaluation, the threshold (if applicable) for all metrics is set to 0.5 to ensure fair comparison.

# 4 Benchmarks

In addition to being fully compatible with existing benchmarks, DeepfakeBench [76] and IMDLBenCo [41], ForensicHub further extends standardization efforts by introducing unified evaluation

Table 2: Summary of ForensicHub datasets. *Pipeline* indicates whether manipulations are manual, typically implying higher quality. *Split* shows if validation and test sets are provided.

| Task | Dataset | Year | Content | Real | Fake | Annotation | Pipeline | Split |
|---|---|---|---|---|---|---|---|---|
| Deepfake | FaceForensics++ [52] | 2019 | Face | 45,388 | 127,209 | Label | Automatic | Train,Test |
| | Celeb-DF-v1 [30] | 2020 | Face | 1,203 | 1,933 | Label | Automatic | Train,Test |
| | Celeb-DF-v2 [30] | 2020 | Face | 5,620 | 10,800 | Label | Automatic | Train,Test |
| | DeepFakeDetection [16] | 2019 | Face | 10,741 | 91,800 | Label | Automatic | Train,Test |
| | DFDC [10] | 2020 | Face | 63,265 | 68,851 | Label | Automatic | Train,Test |
| | DFDCP [11] | 2019 | Face | 5,901 | 11,321 | Label | Automatic | Train,Test |
| | FaceShifter [26] | 2020 | Face | 4,479 | 4,479 | Label | Automatic | Train,Test |
| | UADFV [29] | 2018 | Face | 1,548 | 1,551 | Label | Automatic | Train,Test |
| IMDL | CASIAv1 [12] | 2013 | General | 0 | 920 | Label,Mask | Manual | Train |
| | CASIAv2 [12] | 2013 | General | 7,491 | 5,123 | Label,Mask | Manual | Train |
| | COVERAGE [66] | 2016 | General | 100 | 100 | Label,Mask | Manual | Train |
| | Columbia [22] | 2006 | General | 183 | 180 | Label,Mask | Automatic | Train |
| | IMD2020 [12] | 2020 | General | 414 | 2,010 | Label,Mask | Manual | Train |
| | NIST16 [17] | 2019 | General | 873 | 564 | Label,Mask | Manual | Train |
| | CocoGlide [18] | 2023 | General | 512 | 512 | Label,Mask | Automatic | Train |
| | Autosplice [23] | 2023 | General | 2,273 | 3,621 | Label,Mask | Manual | Train |
| AIGC | DiffusionForensics [65] | 2023 | General | 134,000 | 481,200 | Label | Automatic | Train,Val,Test |
| | GenImage [82] | 2023 | General | 1,331,167 | 1,350,000 | Label | Automatic | Train,Test |
| Document | Doctamper [48] | 2023 | Document | 0 | 170,000 | Mask | Automatic | Train,Test |
| | OSTF [49] | 2025 | Document | 16,405 | 3,685 | Mask | Manual | Train,Test |
| | RealTextManipulation [37] | 2025 | Document | 52,314 | 17,423 | Mask | Automatic+Manual | Train,Test |
| | T-SROIE [64] | 2022 | Document | 21,268 | 4,326 | Mask | Manual | Train,Test |
| | Tampered-IC13 [63] | 2022 | Document | 2,810 | 1,228 | Mask | Manual | Train,Test |

protocols for the AIGC and Document domains—two areas that previously lacked widely accepted benchmarks and codebases. We propose two protocols for two domains to evaluate generalization.

## 4.1 AI Generation Image Detection Benchmark

**Datasets.** In the field of AIGC detection, the challenge in dataset construction usually lies not in obtaining a sufficient quantity of samples, since they can be easily generated using existing models, but in ensuring comprehensive coverage of a wide range of generative models. Therefore, we select only two commonly used public datasets: DiffusionForensics [65] and GenImage [82]. The former contains only diffusion-based generated images, while the latter covers a million-scale dataset constructed from eight SoTA generative models. Models are trained on DiffusionForensics and evaluated on different generative models within GenImage to assess generalization, as detection methods typically already achieve good performance on samples from the same generative model [82]. The detailed data splits are summarized in Table D.1.3.

**Models.** ForensicHub implements five SoTA methods in AIGC detection: Dire [65], DualNet [69], HiFiNet [19], Synthbuster [2], and UnivFD [45], among which Synthbuster has no official open-source code and is fully reimplemented by us. More details about models and training settings can be found in Appendix E.1.

**Results.** Table 3 in green background presents the AUC scores for image-level classification for AIGC benchmark, divided into in-domain results on the test set split of DiffusionForensics [65], and cross-domain results on different generative models and the total set on GenImage [82]. The results show that AIGC SoTAs generally achieve excellent performance on the DiffusionForensics test set, which shares the same source as the training set, and also perform well on datasets composed of diffusion-based generated images like ADM, VQDM, and GLIDE that are similar to the training data. However, the relatively poor generalization to generative models like Midjourney and Wukong highlights areas for improvement and provides guidance for future model development.

## 4.2 Document Image Manipulation Localization Benchmark

**Dataset.** Existing datasets for document image manipulation localization can be broadly categorized into two types: high-fidelity non-sliced datasets, including T-SROIE [64], OSTF [49], TPIC-13 [63], and RTM [37]; and sliced datasets, represented by Doctamper [48]. The primary difference lies in whether the images are preprocessed using patch-wise slicing.

To ensure consistency for downstream evaluation, we adopt the slicing strategy from Doctamper and apply it to the four non-sliced datasets, resulting in a unified format. Each dataset follows its original train/test split. Notably, Doctamper provides one training set and three distinct test

Table 3: AUC scores of image-level detectors. Models are tested in-domain on DiffusionForensics and cross-domain on GenImage sources. $Average_C$ reflects cross-domain performance. In this table, each cell color denotes the model's associated task domain: ▢ Green indicates AIGC models, ▢ Blue denotes IMDL models. Additionally, in other tables ▢ Yellow indicates Deepfake models, ▢ Orange indicates Document models, and ▢ Gray indicates backbone models.

| Method | Within-Domain | Cross-Domain | | | | | | | | |
|---|---|---|---|---|---|---|---|---|---|---|
| | DiffusionForensics | ADM | BigGAN | Midjourney | VQDM | GLIDE | SD V1.4 | SD V1.5 | Wukong | Average_C |
| DualNet [69](APSIPA23) | 1.0000 | 0.9986 | 0.9172 | 0.8495 | 0.9901 | 0.9835 | 0.8127 | 0.8081 | 0.7925 | 0.8902 |
| HiFiNet [19](CVPR23) | 1.0000 | 0.9998 | 0.8407 | 0.7210 | 0.9999 | 0.9991 | 0.6765 | 0.6747 | 0.6675 | 0.8160 |
| Synthbuster [2](OJSP23) | 0.6662 | 0.6226 | 0.8550 | 0.4565 | 0.6308 | 0.7719 | 0.4606 | 0.4614 | 0.3902 | 0.5762 |
| UnivFD [45](CVPR23) | 0.9947 | 0.8400 | 0.9687 | 0.5427 | 0.9663 | 0.9541 | 0.7078 | 0.7072 | 0.6781 | 0.7920 |
| MVSS-Net [5](ICCV21) | 0.9992 | 0.9706 | 0.9520 | 0.6242 | 0.8894 | 0.9380 | 0.7207 | 0.7192 | 0.6089 | 0.7997 |
| CAT-Net [24](IJCV22) | 0.9985 | 0.9391 | 0.9327 | 0.6423 | 0.8226 | 0.7987 | 0.7171 | 0.7142 | 0.6156 | 0.7707 |
| Trufor [18](CVPR23) | 0.9999 | 0.9750 | 0.9316 | 0.7079 | 0.9340 | 0.9144 | 0.8683 | 0.8687 | 0.8018 | 0.8752 |
| IML-ViT [40](Arxiv) | 1.0000 | 0.9594 | 0.9152 | 0.8086 | 0.9577 | 0.9464 | 0.8924 | 0.8877 | 0.8006 | 0.8957 |
| Mesorch [83](AAAI25) | 0.9998 | 0.9754 | 0.9667 | 0.7011 | 0.9331 | 0.9253 | 0.8168 | 0.8179 | 0.7441 | 0.8582 |

Table 4: Binary-F1 scores of document detectors. $Average_D$ is the mean over three Doctamper test dataset, and $Average_{All}$ is averaged across all seven test datasets.

| Model | Doctamper | | | | T-SROIE-train | OSTF-train | TPIC-13-train | RTM-train | Average_All |
|---|---|---|---|---|---|---|---|---|---|
| | DocTamperFCD | DocTamperSCD | DocTamperTest | Average_D | T-SROIE-test | OSTF-test | TPIC-13-test | RTM-test | |
| CAFTB [57](TOMM24) | 0.2917 | 0.3770 | 0.3275 | 0.3321 | 0.9165 | 0.6478 | 0.8394 | 0.2493 | 0.5213 |
| DTD [48](CVPR23) | 0.6856 | 0.7392 | 0.8031 | 0.7426 | 0.9205 | 0.5626 | 0.8341 | 0.1720 | 0.6739 |
| FFDN [6](ECCV24) | 0.8773 | 0.7392 | 0.8212 | 0.8126 | 0.9137 | 0.5586 | 0.8186 | 0.1589 | 0.6982 |
| TIFDM [13](TCE24) | 0.0896 | 0.2572 | 0.2585 | 0.2018 | 0.8942 | 0.5410 | 0.7972 | 0.0591 | 0.4138 |
| MVSS-Net [5](ICCV21) | 0.2066 | 0.3710 | 0.3810 | 0.3195 | 0.8756 | 0.5254 | 0.7864 | 0.1314 | 0.4682 |
| PSCC-Net [34](TCSVT22) | 0.3855 | 0.3931 | 0.4972 | 0.4253 | 0.9305 | 0.5697 | 0.7894 | 0.1971 | 0.5375 |
| Cat-Net [24](IJCV22) | 0.7600 | 0.6405 | 0.7644 | 0.7216 | 0.8726 | 0.5371 | 0.7947 | 0.1174 | 0.6410 |
| IML-ViT [40](Arxiv) | 0.4688 | 0.5117 | 0.4486 | 0.4764 | 0.8731 | 0.5128 | 0.7202 | 0.0920 | 0.5182 |
| Trufor [18](CVPR23) | 0.2613 | 0.3124 | 0.2517 | 0.2751 | 0.9095 | 0.6485 | 0.8197 | 0.2302 | 0.4905 |
| Mesorch [83](AAAI25) | 0.4231 | 0.4586 | 0.4651 | 0.4489 | 0.9049 | 0.4804 | 0.7607 | 0.1314 | 0.5177 |

sets—Doctamper-Test, Doctamper-FCD, and Doctamper-SCD—targeting different manipulation scenarios. The detailed data distribution is summarized in Table D.1.4.

**Model.** We employ two open-source models, DTD [48] and FFDN [6], and reproduce two closed-source models, CATFB [57] and TIFDM [13]. All details are provided in Appendix D.2.4 and E.2.

**Results.** Following the original protocols [48, 6], each detector is trained on its designated training split and evaluated on the corresponding test split. As shown in Table 4, three models consistently achieve top performance: FFDN and DTD, both designed specifically for document forensics, and Cat-Net, an IMDL-based model. Notably, all three methods incorporate JPEG-specific priors, such as DCT coefficients and quantization tables, highlighting the discriminative value of compression artifacts for manipulation localization in document images.

However, this evaluation setting has a key limitation: all models are trained and tested within the same distribution, limiting the assessment of cross-domain generalization. To address this, we introduce a dedicated *Doc Protocol*, where models are trained on Doctamper and evaluated on four other document-level test sets. As shown in Table 5, PSCC-Net demonstrates superior generalization, highlighting the benefit of progressive spatial modeling for Doc manipulation localization.

Table 5: Binary F1 evaluation of models trained only on Doctamper and tested on both within-domain and cross-domain datasets. $Average_W$, $Average_C$, and $Average_{All}$ denote the average performance on Doctamper, external datasets, and all test sets, respectively.

| Model | Within-Domain | | | | Cross-Domin | | | | | Average_All |
|---|---|---|---|---|---|---|---|---|---|---|
| | DocTamperFCD | DocTamperSCD | DocTamperTest | Average_W | T-SROIE | OSTF | TPIC-13 | RTM | Average_C | |
| CAFTB [57](TOMM24) | 0.2917 | 0.3770 | 0.3275 | 0.3321 | 0.2617 | 0.1194 | 0.3007 | 0.0328 | 0.1787 | 0.2444 |
| DTD [48](CVPR23) | 0.6856 | 0.7392 | 0.8031 | 0.7426 | 0.5245 | 0.1241 | 0.2835 | 0.0575 | 0.2474 | 0.4596 |
| FFDN [6](ECCV24) | 0.8773 | 0.7392 | 0.8212 | 0.8126 | 0.5330 | 0.2409 | 0.3572 | 0.0708 | 0.3005 | 0.5199 |
| TIFDM [13](TCE24) | 0.0896 | 0.2572 | 0.2585 | 0.2018 | 0.0582 | 0.0058 | 0.0134 | 0.0176 | 0.0238 | 0.1000 |
| MVSS-Net [5](ICCV23) | 0.2066 | 0.3710 | 0.3810 | 0.3195 | 0.1870 | 0.0373 | 0.1134 | 0.0268 | 0.0911 | 0.1890 |
| PSCC-Net [34](TCSVT22) | 0.3855 | 0.3931 | 0.4972 | 0.4253 | 0.5168 | 0.4414 | 0.5495 | 0.1255 | 0.4083 | 0.4156 |
| Cat-Net [24](IJCV22) | 0.7600 | 0.6405 | 0.7644 | 0.7216 | 0.6085 | 0.1777 | 0.3430 | 0.0630 | 0.2981 | 0.4796 |
| IML-ViT [40](Arxiv) | 0.4688 | 0.5117 | 0.4486 | 0.4764 | 0.4269 | 0.2101 | 0.2563 | 0.0764 | 0.2424 | 0.3427 |
| Trufor [18](CVPR23) | 0.2613 | 0.3124 | 0.2517 | 0.2751 | 0.2126 | 0.0464 | 0.1038 | 0.0342 | 0.0993 | 0.1746 |
| Mesorch [83](AAAI25) | 0.4231 | 0.4586 | 0.4651 | 0.4489 | 0.2937 | 0.1388 | 0.2408 | 0.0405 | 0.1785 | 0.2944 |

Table 6: Comparison of model parameters and FLOPs across representative architectures.

| Model | Params (M) | FLOPs (G) | Model | Params (M) | FLOPs (G) | Model | Params (M) | FLOPs (G) |
|---|---|---|---|---|---|---|---|---|
| Resnet [21] | 44.55 | 20.59 | Capsule-Net [42] | 3.90 | 32.43 | Mesorch [83] | 85.75 | 57.95 |
| Xception [8] | 22.86 | 12.05 | RECCE [4] | 25.83 | 16.18 | DualNet [69] | 7.99 | 66.34 |
| EfficientNet [61] | 19.34 | 4.14 | SPSL [32] | 20.81 | 12.06 | HiFiNet [19] | 6.89 | 145.00 |
| ConvNeXt [36] | 50.22 | 22.74 | MVSS-Net [5] | 147.00 | 83.08 | UnivFD [45] | 428.00 | 156.00 |
| Segformer [70] | 44.07 | 16.29 | Trufor [18] | 68.70 | 112.00 | DTD [48] | 67.07 | 272.00 |
| Swin [35] | 49.61 | 28.65 | IML-ViT [40] | 91.78 | 80.37 | FFDN [6] | 89.20 | 453.00 |

Table 7: Cross-domain AUC evaluation of models trained under *IFF-Protocol*.

| Method | Deepfake | | | IMDL | | | AIGC | | Document | | | Average |
|---|---|---|---|---|---|---|---|---|---|---|---|---|
| | FF-c40 | CDFv2 | DFD | Columbia | IMD2020 | Autosplice | DF | GenImage | T-SROIE | OSTF | RTM | |
| Resnet [21](CVPR16) | 0.681 | 0.730 | 0.793 | 0.482 | 0.533 | 0.738 | 0.619 | 0.797 | 0.951 | 0.681 | 0.662 | 0.697 |
| Xception [8](CVPR17) | 0.728 | 0.719 | 0.870 | 0.465 | 0.537 | 0.756 | 0.757 | 0.980 | 0.966 | 0.762 | 0.734 | 0.752 |
| EfficientNet [61](IJCML19) | 0.504 | 0.535 | 0.517 | 0.623 | 0.506 | 0.483 | 0.544 | 0.597 | 0.884 | 0.581 | 0.512 | 0.571 |
| Segformer [70](NIPS21) | 0.691 | 0.748 | 0.862 | 0.409 | 0.562 | 0.824 | 0.805 | 0.998 | 0.980 | 0.866 | 0.736 | 0.771 |
| Swin [35](ICCV21) | 0.771 | 0.746 | 0.901 | 0.636 | 0.631 | 0.864 | 0.915 | 0.999 | 0.990 | 0.856 | 0.758 | 0.824 |
| ConvNext [36](CVPR22) | 0.794 | 0.784 | 0.911 | 0.625 | 0.598 | 0.825 | 0.895 | 1.000 | 0.994 | 0.849 | 0.762 | 0.822 |
| Capsule-Net [42](ICASSP19) | 0.613 | 0.660 | 0.699 | 0.330 | 0.527 | 0.745 | 0.546 | 0.971 | 0.946 | 0.704 | 0.670 | 0.674 |
| RECCE [4](CVPR22) | 0.634 | 0.602 | 0.727 | 0.506 | 0.492 | 0.642 | 0.684 | 0.906 | 0.542 | 0.688 | 0.555 | 0.634 |
| SPSL [32](CVPR21) | 0.730 | 0.726 | 0.876 | 0.419 | 0.545 | 0.759 | 0.770 | 0.987 | 0.972 | 0.769 | 0.738 | 0.754 |
| Sia [60](ECCV22) | 0.629 | 0.584 | 0.671 | 0.653 | 0.483 | 0.626 | 0.593 | 0.748 | 0.610 | 0.677 | 0.574 | 0.622 |
| Effort [73](IJCML25) | 0.805 | 0.846 | 0.930 | 0.979 | 0.861 | 0.943 | 0.930 | 0.992 | 0.960 | 0.834 | 0.732 | 0.892 |
| MVSS-Net [5](ICCV21) | 0.713 | 0.700 | 0.857 | 0.298 | 0.539 | 0.795 | 0.671 | 0.994 | 0.978 | 0.806 | 0.741 | 0.736 |
| Trufor [18](CVPR23) | 0.642 | 0.698 | 0.832 | 0.306 | 0.564 | 0.808 | 0.726 | 0.996 | 0.979 | 0.805 | 0.732 | 0.735 |
| IML-ViT [40](Arxiv) | 0.750 | 0.726 | 0.851 | 0.483 | 0.556 | 0.819 | 0.627 | 0.991 | 0.972 | 0.800 | 0.703 | 0.753 |
| Mesorch [83](AAAI25) | 0.767 | 0.814 | 0.867 | 0.285 | 0.570 | 0.773 | 0.629 | 0.996 | 0.982 | 0.819 | 0.739 | 0.749 |
| DualNet [69](APSIPA23) | 0.637 | 0.552 | 0.540 | 0.268 | 0.517 | 0.748 | 0.899 | 0.988 | 0.935 | 0.658 | 0.657 | 0.673 |
| HiFiNet [19](CVPR23) | 0.587 | 0.611 | 0.648 | 0.745 | 0.534 | 0.677 | 0.575 | 0.756 | 0.937 | 0.663 | 0.615 | 0.668 |
| UnivFD [45](CVPR23) | 0.690 | 0.671 | 0.798 | 0.886 | 0.786 | 0.785 | 0.742 | 0.813 | 0.938 | 0.684 | 0.569 | 0.760 |
| FatFormer [33](CVPR24) | 0.842 | 0.770 | 0.866 | 0.199 | 0.585 | 0.784 | 0.941 | 0.999 | 0.983 | 0.806 | 0.751 | 0.758 |
| CO-SPY [7](CVPR25) | 0.819 | 0.780 | 0.875 | 0.460 | 0.716 | 0.779 | 0.940 | 0.989 | 0.969 | 0.836 | 0.748 | 0.829 |
| DTD [48](CVPR23) | 0.498 | 0.520 | 0.490 | 0.679 | 0.498 | 0.506 | 0.457 | 0.499 | 0.748 | 0.595 | 0.496 | 0.544 |
| FFDN [6](ECCV24) | 0.714 | 0.699 | 0.871 | 0.553 | 0.624 | 0.927 | 0.999 | 1.000 | 0.997 | 0.893 | 0.782 | 0.824 |

## 5 Image Forensic Fusion Protocol

**Protocol.** To explore the performance of different models under a unified forensic protocol, we implement an *image forensic fusion protocol (IFF-Protocol)* inspired by CAT-Net's training data construction strategy. The *IFF-Protocol* defines the training set as a combination of Deepfake, IMDL, AIGC, and Document data, where each training epoch samples an equal number of instances from each domain at random. During training, we select FaceForensics++ [52] from Deepfake, CASIAv2 [12] from IMDL, GenImage [82] from AIGC, and OSTF [49], RealTextManipulation [37], T-SROIE [64], and Tampered-IC13 [63] from the Document. We use the smallest dataset, CASIAv2 with 12,641 samples, as the sampling number for each epoch. During testing, we evaluate the models directly on datasets from different domains without fine-tuning.

**Implementation Details.** We resize images to *256×256* (except UnivFD, DTD and FFDN, see Appendix F for details) and apply only basic data augmentations, including flipping, brightness and contrast adjustment, compression, and Gaussian blur. All models are trained for 20 epochs using a cosine decay learning rate schedule, decreasing from $1e-4$ to $1e-5$. For models that output masks (IMDL and Doc), we apply max pooling to the final-layer feature maps to obtain a predicted label and compute the loss using only the label.

**Model Efficiency.** We test the parameters and FLOPs of the backbones and SoTAs of each domain in Table 6. It can be observed that model efficiency is often related to the task's application scenario. For example, Deepfake models are typically lightweight to support real-time video detection, while IMDL models, which focus on pixel-level classification, often adopt more complex and heavier architectures. These efficiency preferences can influence the experimental results under the *IFF-Protocol*.

**Benchmark Result.** Table 7 shows the AUC scores of backbones and domain-specific SoTA methods on datasets from four domains under the *IFF-Protocol*, in which DFD refers to DeepFakeDetection [16], DF refers to DiffusionForensics [65], and RTM refers to RealTextManipulation [37]. We provide detailed results in Appendix F.

The results show that surprisingly, visual backbones such as ConvNeXt [36] and Swin Transformer [35] outperform almost all domain-specific SoTA methods, indicating that backbones demonstrate greater potential when trained on more unified fake images. Meanwhile, domain-specific SoTAs

Table 8: Mean AUC differences between extractor-enhanced models and their plain counterparts. Positive values (red) indicate gains; negative (blue) indicate drops.

| Task | Extractor = Sobel [5] | | | | | | Extractor = Bayar[3] | | | | | |
|---|---|---|---|---|---|---|---|---|---|---|---|---|
| | ResNet | Xception | EfficientNet | SwinTransformer | SegFormer | ConvNext | ResNet | Xception | EfficientNet | SwinTransformer | SegFormer | ConvNext |
| AIGC | -0.0446 | -0.0513 | 0.0359 | -0.0983 | -0.1047 | -0.1374 | -0.0582 | -0.0258 | -0.0105 | -0.0709 | -0.0705 | -0.0672 |
| Deepfake | -0.0887 | -0.0163 | 0.0273 | -0.1319 | -0.1262 | -0.1657 | -0.0749 | -0.0103 | 0.0088 | -0.0963 | -0.0973 | -0.0878 |
| Document | -0.0450 | -0.0268 | 0.0063 | -0.0950 | -0.0815 | -0.1228 | -0.0333 | -0.0058 | 0.0025 | -0.0760 | -0.0653 | -0.0885 |
| IMDL | 0.0093 | -0.0203 | 0.0058 | -0.0838 | -0.0642 | -0.0687 | -0.0118 | -0.0227 | 0.0085 | -0.0938 | -0.0735 | -0.0913 |
| | Extractor = FPH [48] | | | | | | Extractor = DCT [1] | | | | | |
| | ResNet | Xception | EfficientNet | SwinTransformer | SegFormer | ConvNext | ResNet | Xception | EfficientNet | SwinTransformer | SegFormer | ConvNext |
| AIGC | -0.1314 | -0.1060 | 0.1425 | -0.0940 | -0.1018 | -0.5071 | -0.0173 | -0.0009 | 0.0081 | -0.0429 | -0.0339 | -0.0704 |
| Deepfake | -0.1380 | -0.1003 | 0.0596 | -0.0585 | -0.1404 | -0.3443 | 0.0055 | -0.0062 | 0.0108 | -0.0078 | -0.0153 | -0.0993 |
| Document | -0.0525 | -0.0645 | 0.0158 | -0.0690 | -0.0965 | -0.2015 | -0.0088 | 0.0000 | 0.0063 | -0.0493 | -0.0213 | -0.0820 |
| IMDL | -0.0128 | -0.0240 | 0.0367 | -0.0917 | -0.0933 | -0.0848 | 0.0102 | 0.0067 | 0.0087 | -0.0565 | -0.0493 | -0.0983 |

do not necessarily retain their superiority within their own tasks. For instance, UnivFD [45], a CLIP-based fine-tuned model for AIGC detection, demonstrates strong performance on the IMD2020 [44] from IMDL, revealing valuable insights into the transferability of cross-task methods.

From a task perspective, although IMDL target shifts from pixel-level to image-level classification, it remains challenging due to significant distribution differences across datasets in terms of size and manipulation types. In contrast, AIGC benefits from training on sufficient data from diverse generative models, resulting in higher detection accuracy. This observation reminds us that it is essential not only to include a comprehensive range of manipulation types in the training data but also to focus on enhancing the generalization ability of models.

## 6 Experiments

Based on ForensicHub, we conduct cross-task experiments, which have been less explored in previous research. The similarities and differences among detection methods across different tasks lead us to the following questions: **1)** *Are low-level feature extractors effective across all tasks?* **2)** *Do detection methods from one task remain effective when transferred to another task?* We answer the above questions through extensive experiments.

### 6.1 Effectiveness of Low-Level Feature Extractors

Since each domain has proposed specific feature extractors, to explore their effectiveness under a unified domain, we conduct experiments using 6 backbones combined with 4 different extractors in the shallow layer under the aforementioned *IFF-Protocol* setting. The extractors are BayarConv [3] for noise aritfacts, Sobel [5] for edge arifacts, DCT [1] and FPH [48] for frequency artifacts. Details of each extractor can be found in Appendix G.1.

Results in Table 8 show the AUC differences between versions of each backbone using the four different feature extractors and those without, averaged across all test datasets for each task. All backbones except EfficientNet [61] show performance drops after using feature extractors, indicating that under the *IFF-Protocol*, where training data includes sufficient manipulation types and image quantity, models do not rely on the additional information provided by feature extractors. However, due to its lightweight nature, EfficientNet still benefits from the use of feature extractors. The results suggest that feature extractors may only be beneficial for detection on small-scale datasets, with limited manipulation types, or when using lightweight models. Details of each domain test datasets AUC scores can be found in Appendix G.2.

### 6.2 Transferability of Task-Specific Detectors

#### 6.2.1 Cross-Evaluation Between IMDL and Document Benchmarks

Current Document-level detectors' input–output formats are fully compatible with those of Image Manipulation Detection and Localization models. Leveraging this consistency, we perform a bidirectional evaluation: IMDL detectors are tested on the Document benchmark, and Document detectors are tested on the IMDL benchmark. This cross-testing enlarges the effective model pool for each benchmark and allows us to probe detector generality beyond their original task scopes.

Table 9: Pixel-level binary F1 evaluation on IMDL benchmarks for document-trained detectors.

| Model | CASIAv1 | COVERAGE | Columbia | IMD2020 | NIST16 | Average |
|---|---|---|---|---|---|---|
| CAFTB [57](TOMM24) | 0.6234 | 0.2557 | 0.6731 | 0.3526 | 0.3770 | 0.4564 |
| DTD [48](CVPR23) | 0.3535 | 0.1482 | 0.6470 | 0.1749 | 0.1811 | 0.3009 |
| FFDN [6](ECCV24) | 0.5012 | 0.2670 | 0.6085 | 0.2516 | 0.2957 | 0.3848 |
| TIFDM [13](TCE24) | 0.3675 | 0.2087 | 0.3572 | 0.1634 | 0.2333 | 0.2660 |

Table 10: Image-level AUC evaluation of IMDL-based detectors trained on FF++-c23 and tested across both within-domain and cross-domain deepfake benchmarks.

| Model | Within Domain Evaluation | | | | | | | Cross Domain Evaluation | | | | | | | |
|---|---|---|---|---|---|---|---|---|---|---|---|---|---|---|---|
| | FF-c23 | FF-c40 | FF-DF | FF-F2F | FF-FS | FF-NT | Avg. | CDFv1 | CDFv2 | DFD | DFDC | DFDCP | Fsh | UADFV | Avg. |
| CAT-Net [24](IJCV22) | 0.9855 | 0.8351 | 0.9946 | 0.9891 | 0.9905 | 0.9689 | 0.9606 | 0.7489 | 0.7511 | 0.8086 | 0.7196 | 0.7043 | 0.6626 | 0.9282 | 0.7605 |
| IML-ViT [40](Arxiv) | 0.9614 | 0.8483 | 0.9810 | 0.9728 | 0.9793 | 0.9206 | 0.9439 | 0.7250 | 0.7419 | 0.7770 | 0.7516 | 0.7237 | 0.6149 | 0.8718 | 0.7437 |
| Mesorch [83](AAAI25) | 0.9815 | 0.8445 | 0.9889 | 0.9890 | 0.9859 | 0.9636 | 0.9589 | 0.7900 | 0.7649 | 0.8468 | 0.7806 | 0.7345 | 0.6638 | 0.9623 | 0.7918 |
| MVSS-Net [5](ICCV21) | 0.9723 | 0.8140 | 0.9779 | 0.9865 | 0.9868 | 0.9458 | 0.9472 | 0.6969 | 0.6839 | 0.7749 | 0.6764 | 0.6785 | 0.5841 | 0.8871 | 0.7117 |
| PSCC-Net [34](TCSVT22) | 0.7147 | 0.6536 | 0.7751 | 0.6414 | 0.6460 | 0.6065 | 0.6729 | 0.4602 | 0.5898 | 0.5565 | 0.5945 | 0.5845 | 0.6574 | 0.8216 | 0.6092 |
| Trufor [18](CVPR23) | 0.7303 | 0.6848 | 0.8087 | 0.7687 | 0.7084 | 0.6326 | 0.7223 | 0.6551 | 0.6115 | 0.5908 | 0.6017 | 0.5749 | 0.5977 | 0.9526 | 0.6549 |

**IMDL → Document.**  Table 4 reports the *within-domain* results obtained under the original Document benchmark split, whereas Table 5 presents the *cross-domain* scores produced by our newly introduced generalization protocol. Across both settings, IMDL detectors demonstrate strong competitiveness in the document forensics scenario. In the conventional split, the Cat-Net [24, 48, 6] family achieves the best average F1, confirming the merit of its hierarchical "cat-net" paradigm. Under the more challenging cross-domain evaluation, PSCC-Net [34] displays markedly better generalization, suggesting that progressive spatial modeling captures cues for document manipulation localization. We expect future work to further investigate the underlying mechanisms behind PSCC-Net.

**Document → IMDL.**  Following the MVSS training protocol [41], all Document-oriented models are trained on the CASIAv2 dataset [12] and evaluated on five standard IMDL test sets. As shown in Table 9, the dual-branch architecture of *CAFTB* [57] achieves the best overall performance among Document models when transferred to IMDL tasks—an outcome that aligns with the design philosophy of the current SoTA model *Mesorch* [83], which also emphasizes dual-branch learning.

### 6.2.2 Extending IMDL Detectors to AIGC and Deepfake Benchmarks

IMDL models are designed to produce both pixel-level masks and image-level labels, with most architectures incorporating classification heads alongside segmentation branches. This dual-output design enables direct application to tasks like AIGC and Deepfake detection. For models without label heads, image-level scores are obtained via max-pooling over the predicted masks.

**IMDL → AIGC.**  We fine-tune representative IMDL detectors on the training split of the AIGC benchmark and report cross-generator performance in Table 3. The training settings and other configurations are consistent with those used in the previously mentioned AIGC benchmark setup. The results show that techniques from IMDL, such as noise print (TruFor) and multiscale analysis (IML-ViT), remain effective for AIGC detection.

**IMDL → Deepfake.**  We train each IMDL detector on the FF++-c23 training split and evaluate on all remaining deep-fake test sets; the scores are given in Table 10. When compared to all baselines in the deepfakebench [76], **Cat-Net** attains the best performance in the *within-domain* setting, while **Mesorch** achieves the highest average accuracy in the *cross-domain* evaluation, establishing new state-of-the-art results in both regimes.

### 6.3 Grad-CAM Visualization

We use Grad-CAM to visualize the heatmaps of models from the four domains (Capsule-Net (Deepfake), MVSS-Net (IMDL), UnivFD (AIGC), DTD (Doc)) on datasets from each domain, aiming to explore their attention regions, as shown in Figure 2. We use Grad-CAM to visualize the heatmaps of models from the four domains. Models from different domains show both common and distinct attention patterns. For Deepfake, CapsuleNet focuses on specific facial features, while MVSS-Net attends to larger areas. For Doc, CapsuleNet, MVSS-Net, and UnivFD capture overall tampered regions, whereas DTD targets subtle traces like edges and curves of characters.

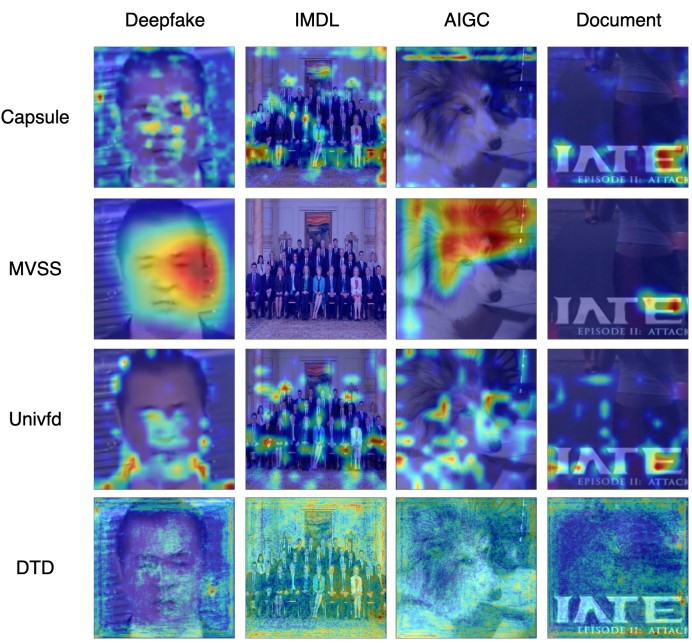

Figure 2: Grad-CAM visualization (zoomed in for better visualization).

## 7 Conclusion

This paper proposes ForensicHub, the first unified benchmark and codebase for all-domain fake image detection and localization. It adapts existing benchmarks and extends to other domains. Based on the extensive cross-domain experiments, we summarize 8 key actionable insights for future research:

1) In Doc, PSCC-Net exhibits strong generalization, while Cat-Net effectively adapts to synthetic manipulations, offering valuable Doc model designs. 2) In IMDL, parallel architecture models like CAFTB and Mesorch achieve leading performance, suggesting the effectiveness of multi-branch modeling. 3) Frequency-strategy models like CAT-Net and Mesorch consistently perform well, highlighting the potential of frequency features for FIDL. 4) Less-explored backbones like ConvNeXt and Swin Transformer outperform nearly all domain SoTAs under *IFF-Protocol*. 5) Shallow concatenation of feature extractors tends to negatively impact performance when the dataset is large and contains a wide variety of manipulation types, while lightweight models such as EfficientNet can benefit from this approach. 6) Current AIGC and Doc evaluations often neglect generalization, leading to overestimated performance. We recommend our proposed AIGC and Doc protocols for future work, which explicitly encourage generalization-aware model design. 7) Existing AIGC and Deepfake datasets are often too simple and lack diversity, limiting meaningful comparisons. Future benchmarks should aim for greater complexity and realism. 8) For all-domain scenarios, we recommend our *IFF-Protocol* to enable more comprehensive evaluation.

In conclusion, ForensicHub represents an important step toward breaking down domain silos across four fields, offering new insights into FIDL future research across model architecture, dataset characteristics, and evaluation standards.

## 8 Acknowledgment

This research was supported by the Sichuan Province Major Special Project (2024ZDZX0001-3), Sichuan Province Natural Science Foundation (Grant No.2024YFHZ0355), and the Science and Technology Development Fund, Macau SAR, under Grant 0193/2023/RIA3 and 0079/2025/AFJ. The authors would like to give special thanks to Dr. Wentao Feng for the workplace, computation power, and physical infrastructure support.

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

# A   Limitations

Certain experiment limitations still remain. For example, in our study of feature extractors, we only apply shallow fusion of features. More advanced fusion strategies that can better exploit the proposed features remain to be explored. ForensicHub will gradually improve and expand its experiments in future versions.

# B   Author Contributions

The author contributions are: **Bo Du:** framework design, construction of AIGC benchmark, IFF-Protocol, feature extractors, cross-domain evaluation, and manuscript writing. **Xuekang Zhu:** framework design, construction of benchmark adapters, Document benchmark, cross-domain evaluation, and manuscript writing. **Xiaochen Ma:** framework design, framework code optimization and manuscript writing. **Chenfan Qu**: construction of Document benchmark and manuscript writing. **Kaiwen Feng:** construction of IFF-Protocol, feature extractors and manuscript writing. **Zhe Yang:** construction of AIGC benchmark. **Chi-Man Pun**: project advising. **Jian Liu:** general project advising. **Jizhe Zhou:** project supervisor and manuscript writing.

# C   Task Definitions and Detection Paradigms

To provide contextual understanding of the forensic domains included in our benchmark, we summarize the goals, characteristics, and representative modeling approaches for each task below.

## C.1   Deepfake Detection

Deepfake detection aims to identify whether faces in an image have been manipulated, typically formulated as an image-level binary classification task. According to [76], current methods fall into three categories: naive, spatial, and frequency detectors. These approaches primarily target artifacts specific to facial manipulation, such as biological signals, spatial inconsistencies, frequency abnormalities, and auto-learned clues. It's important to note that artifacts characteristic of Deepfake images often differ from those found in other types of image manipulation.

## C.2   Image Manipulation Detection and Localization

IMDL task requires two types of assignments [41]: image-level detection to determine whether manipulation has occurred, and pixel-level localization to identify the manipulated regions. IMDL models are typically composed of a backbone and a low-level feature extractor to capture artifacts left by manipulation, such as edge artifacts [5, 40], frequency artifacts [24], and noise artifacts [18, 5, 83]. IMDL models are generally designed to detect manipulations in natural images rather than targeting specific types of tampering, such as facial forgeries.

## C.3   AI-Generated Image Detection

AI-Generated Image Detection focuses on identifying whether an image is generated by generative models, performing binary classification at the image level only. Existing classifiers typically detect AIGC images by leveraging artifacts that differentiate them from real images, such as discrepancies in spatial feature space [45, 19], frequency inconsistency [46, 69, 2], and fingerprints left by specific generative models like diffusion models [65, 56, 39]. As a rapidly evolving technology, AIGC presents challenges to detection methods due to the artifacts left by deep generative models, which differ significantly from those found in traditional manual manipulations.

## C.4   Document Image Manipulation Localization

Document Image Manipulation Localization focuses on identifying tampered text on images. Tampered text regions are usually small in size, with subtle appearance anomalies and fewer edge artifacts, due to consistent backgrounds and fonts [57, 53]. Consequently, methods designed for detecting forgeries in natural and face images usually do not perform well when applied directly to this task [13, 64].

To overcome this difficulty, recent studies propose to model the block artifact grids [48] or the texture differences [63], etc. Despite progress, accurately detecting forged text against elaborate tampering processes, advanced text editing models, and diverse image styles remains an open challenge [37, 49].

# D  Details of ForensicHub Construction

## D.1  Datasets

### D.1.1  Deepfake

Most Deepfake datasets are video-based. Following the protocol of DeepfakeBench [76], we extract 32 equally spaced frames from each video to form the image-based datasets used in our experiments.

**FaceForensics++.**  FaceForensics++ (FF++) [52] is the most widely used benchmark for Deepfake detection. It provides real and fake data generated using four manipulation methods: DeepFakes (FF-DF), Face2Face (FF-F2F), FaceSwap (FF-FS), and NeuralTextures (FF-NT). These four subsets share the same test real images but differ in the fake generation methods. The full dataset contains 27,472 real and 109,800 fake images.

The training set includes 22,993 real and 91,891 fake images. The test set contains 4,479 real images, shared across four manipulation subsets, with fake image counts of 4,473 (FF-DF), 4,480 (FF-F2F), 4,477 (FF-FS), and 4,479 (FF-NT), respectively.

These subsets are designed to assess how detection models perform against different generation types.

**Celeb-DF-v1.**  Celeb-DF-v1 [30] was released in 2020 with improved realism over early datasets. It includes 7,946 real and 25,362 fake training images, and 1,203 real and 1,933 fake test images.

**Celeb-DF-v2.**  Celeb-DF-v2 [30] expands on v1 in both quality and size. The training set contains 9,524 real and 179,777 fake images. The test set includes 5,620 real and 10,800 fake images.

**DeepFakeDetection.**  The DeepFakeDetection (DFD) [16] dataset, released by Google and Jigsaw, provides 10,741 real and 91,800 fake images for both training and testing. It is widely used for large-scale pretraining and evaluation.

**DFDCP.**  The DFDCP [11] dataset introduces post-compressed versions of fake images to simulate real-world distortions. The training set contains 22,425 real and 103,631 fake images. The test set includes 5,901 real and 11,321 fake images.

**DFDC.**  The Deepfake Detection Challenge (DFDC) [10] dataset, provided by Facebook, contains only a test set with 63,265 real and 68,851 fake images. The training set is not publicly available.

**FaceShifter.**  The FaceShifter [25] dataset offers 22,993 real and 22,968 fake training images, and 4,479 real and 4,479 fake images for testing. It is typically used to assess model generalization to unseen generation techniques.

**UADFV.**  UADFV [29] is one of the earliest Deepfake datasets. Both the training and test sets contain 1,548 real and 1,551 fake images. Due to its small size and early generation style, it is mostly used for cross-dataset evaluation.

### D.1.2  IMDL

Due to the difficulty of annotation, IMDL datasets are typically small in scale. Information on datasets such as CASIA [12], Columbia [22], COVERAGE [66], IMD2020 [44], and NIST16 [17] can be found in IMDLBenCo [41]. Notably, considering the recent rise of AI-based partial image inpainting manipulations, we include two inpainting datasets generated using deep generative models: CocoGlide [18] and Autosplice [23]. CocoGlide and AutoSplice include 512 and 3621 images edited by GLIDE diffusion model and DALL-E2, respectively.

### D.1.3 AIGC

**DiffusionForensics.** DiffusionForensics [65] is a dataset constructed to facilitate the evaluation of detectors targeting diffusion-generated images. It comprises real and synthetic images across three representative domains: LSUN-Bedroom, ImageNet, and CelebA-HQ. The dataset includes outputs from a variety of diffusion models, covering unconditional, class-conditional, and text-to-image generation paradigms. For each image, the dataset provides a triplet: the source image, its reconstruction, and the corresponding DIRE image, enabling more detailed forensic analysis. The design of DiffusionForensics supports both training and testing, with subsets carefully split for each purpose. By encompassing a wide range of generative models and image domains, it serves as a comprehensive benchmark for assessing the generalization and robustness of diffusion image detectors.

**GenImage.** GenImage [82] is a large-scale dataset developed to advance the detection of AI-generated images. It contains over one million pairs of synthetic and real images, covering a wide range of image categories. The synthetic images are produced using state-of-the-art generative models, including advanced diffusion models and GANs. GenImage includes the generative models of ADM, BigGAN, Midjourney, VQDM, GLIDE, Stable Diffusion V1.4, Stable Diffusion V1.5, Wukong. Each generative model generates nearly the same numbers of images (approximately 168750), with a total number of 1,350,000 of fake images. GenImage enables the evaluation of detectors under realistic conditions through two tasks: cross-generator classification, which assesses generalization across different generative models, and degraded image classification, which tests robustness to image quality degradation such as compression, blurring, and low resolution. By combining scale, diversity, and challenging evaluation settings, GenImage provides a comprehensive benchmark for developing reliable fake image detectors.

### D.1.4 Document

**DocTamper.** DocTamper [48] was introduced in 2023 and has become the most widely used dataset for document tampering localization. It contains fully synthetic manipulations on various photographed documents, such as contracts, receipts, invoices, and books. The tampering types include copy-move, splicing, and print-based edits. All images have been preprocessed by cropping to $512 \times 512$ resolution, and the corresponding pixel-level masks are cropped accordingly. The training set contains 120,000 fake images, while the test set is split into three subsets: DocTamper-Test (30,000 fake), DocTamper-FCD (2,000 fake), and DocTamper-SCD (18,000 fake). Clean images are not included.

**T-SROIE.** Released in 2022, T-SROIE [64] is the first dataset to localize AIGC-style tampering in scanned receipts using a modern IML approach. It contains text tampered by SR-Net [67] and was originally provided as high-resolution uncropped images. To ensure consistency with DocTamper, we apply the same cropping strategy to resize all images to $512 \times 512$, and crop the corresponding pixel-level masks in the same manner. After cropping, the training set consists of 12,769 real and 2,747 fake images; the test set contains 8,499 real and 1,579 fake images.

**RTM.** RTM [37] was introduced in 2025 and includes both synthetic and manually manipulated document images. The dataset covers a wide range of manipulation types, including copy-move, splicing, print, and erasure, across diverse document types such as scanned forms. The original high-resolution images are not cropped, so we apply the DocTamper-style cropping strategy to obtain $512 \times 512$ resolution images, along with their aligned masks. After cropping, the RealTextManipulation-Test set includes 22,334 real and 3,444 fake samples.

**OSTF.** OSTF [49], proposed in 2025, contains natural scene texts tampered by eight different AIGC-based text editing models. It focues on evaluating model generalization ability across unseen text tampering models and unseen image styles. Since the original images are high-resolution and unaligned, we perform $512 \times 512$ cropping using the DocTamper protocol, and apply the same transformation to the associated masks. The resulting training set includes 1,729 real and 639 fake samples; the test set includes 14,676 real and 3,046 fake samples.

**Tampered-IC13.** Tampered-IC13, released in 2022, contains naturally captured scene texts tampered by the AIGC text editing model SR-Net [67]. It also lacks predefined cropping, so we apply the DocTamper-style image and mask cropping to $512 \times 512$. After preprocessing, the training set includes 1,729 real and 639 fake images; the test set includes 1,081 real and 589 fake images.

## D.2 Models

### D.2.1 Deepfake

For Deepfake detection, we design an adapter to directly align with the 27 image-based detectors provided in DeepfakeBench [76]. These detectors cover diverse architectures and training settings. For full details, we refer to the official DeepfakeBench documentation.

### D.2.2 IMDL

We adapt all nine detection models from IMDLBenCo [41] via adapters. For detailed information on these models, please refer to the official IMDLBenCo documentation.

### D.2.3 AIGC

**Dire.** Dire [65] is a novel approach designed to detect diffusion-generated images by leveraging a unique image representation called Diffusion Reconstruction Error (DIRE). Unlike existing detectors, which often struggle to distinguish between real and diffusion-generated images, DIRE measures the reconstruction error between an input image and its counterpart reconstructed by a pre-trained diffusion model. It has been observed that while diffusion-generated images can be effectively reconstructed by a diffusion model, real images cannot, making DIRE a valuable tool for distinguishing between the two. DIRE is robust to various perturbations and generalizes well across different diffusion models, even those not seen during training. Extensive experiments on a comprehensive benchmark dataset demonstrate that DIRE outperforms previous detection methods in identifying AI-generated images, establishing it as a powerful tool for diffusion-based image forensics.

**DualNet.** DualNet [69] is a novel detection method developed to address the challenges posed by AI Generated Content (AIGC), particularly text-to-image models like DALL·E2 and DreamStudio. Unlike traditional computer-generated graphics (CG), AIGC images are inherently more deceptive and require less human intervention, making conventional CG detection methods inadequate. To improve detection, DualNet employs a robust dual-stream network consisting of a residual stream and a content stream. The residual stream uses the Spatial Rich Model (SRM) to extract texture information from images, while the content stream captures low-frequency forged traces, providing complementary insights. These two streams are connected through a cross multi-head attention mechanism to enhance information exchange. Extensive experiments on two text-to-image databases and traditional CG benchmarks, such as SPL2018 and DsTok, demonstrate that DualNet consistently outperforms existing detection methods across a range of image resolutions, showing superior robustness and generalization capabilities.

**HiFiNet.** HiFiNet [19] is a novel framework designed to address the challenges of image forgery detection and localization (IFDL), particularly when distinguishing between images generated by CNN-based synthesis and image-editing techniques. Due to the significant differences in forgery attributes between these domains, traditional methods struggle to provide a unified solution. HiFiNet tackles this issue by employing a hierarchical fine-grained approach for IFDL representation learning. The method first represents forgery attributes with multiple labels at different levels and performs fine-grained classification using the hierarchical dependencies between them. This encourages the model to learn both comprehensive features and the inherent hierarchical nature of various forgery attributes, improving the detection and localization performance. HiFiNet consists of three key components: a multi-branch feature extractor that classifies forgery attributes at different levels, and localization and classification modules that segment pixel-level forgery regions and detect image-level forgery, respectively. The effectiveness of HiFiNet is demonstrated through experiments on seven different benchmarks, showing significant improvements in both IFDL and forgery attribute classification tasks.

**Synthbuster.** Synthbuster [2] is a detection method specifically designed to identify images generated by diffusion models, a type of AI-based generative technique that has gained popularity due to its ability to produce photo-realistic images from simple text prompts. While older detection methods targeting Generative Adversarial Networks (GANs) exist, they are insufficient for detecting images from advanced diffusion models. Synthbuster addresses this gap by focusing on the unique frequency artifacts left behind during the diffusion process. The method uses spectral analysis of the Fourier transform of residual images to highlight these artifacts, enabling the distinction between real and synthetic images. Synthbuster demonstrates strong detection capabilities even in the presence of mild JPEG compression and generalizes effectively to unseen models. This novel approach aims to enhance forensic techniques for detecting AI-generated images and encourages further research into this emerging field.

**UnivFD.** UnivFD [45] is a novel approach designed to address the growing need for general-purpose fake image detectors, particularly in the face of rapidly evolving generative models. Traditional methods, which rely on training deep networks for real-vs-fake classification, struggle to detect images generated by newer models when trained only on older generative models like GANs. Analysis reveals that such classifiers become asymmetrically tuned, with the "real" class effectively acting as a catch-all for any image that isn't fake, leading to poor performance when confronted with images generated by models not seen during training. To overcome this, UnivFD introduces a novel strategy: performing real-vs-fake classification without explicit training, using a feature space that is not designed to distinguish between real and fake images. By leveraging the feature space of large pretrained vision-language models and applying simple methods like nearest neighbor and linear probing, UnivFD achieves surprisingly strong generalization.

### D.2.4 Document

**PS-Net.** We reproduce PS-Net [53], a tampering localization model that refines both the input and output stages to enhance detection performance. At the input level, a Multi-View Enhancement (MVE) module fuses RGB, noise residual, and texture features to capture richer tampering traces. At the output level, Progressive Supervision (PS) applies multi-scale BCE losses to exploit hierarchical localization cues, while a Detection Assistance (DA) module introduces KL loss to align detection and localization branches. PS-Net demonstrates strong performance on DocTamper, effectively combining fine-grained supervision and global consistency.

**CAFTB-Net.** We reproduce CAFTB-Net [57], a dual-branch network designed for document forgery localization in complex and noisy environments. It consists of a Spatial Information Extraction Branch (SIEB) and a Noise Feature Extraction Branch (NFEB), with the latter leveraging a Spatial Rich Model (SRM) filter to extract tampering cues. A Cross-Attention Fusion Module (CAFM) integrates both branches to enhance localization. CAFTB-Net achieves strong performance across benchmarks, particularly in detecting subtle and diverse manipulations.

**TIFDM.** We reproduce TIFDM [13], which performs document forgery localization by modeling spatial. It processes RGB and uses attention mechanisms and a multi-scale decoder to improve localization. TIFDM shows robust generalization to mixed tampering types, including splicing, erasure, and generative edits.

**DTD.** DTD (Document Tampering Detector) [48] introduces a multi-modality framework for detecting tampered text in document images. It integrates both RGB visual features and frequency cues extracted from JPEG compression artifacts via a dedicated Frequency Perception Head (FPH). A Swin-Transformer encoder combined with a Multi-view Iterative Decoder (MID) enables the model to capture subtle and dispersed tampering signals. Furthermore, DTD incorporates Curriculum Learning for Tampering Detection (CLTD), training the model in an easy-to-hard strategy to enhance robustness against varying compression levels. Extensive evaluations on DocTamper and T-SROIE datasets show that DTD achieves state-of-the-art performance, particularly in scenarios with heavy JPEG compression and complex document layouts.

**FFDN.** FFDN (Feature Fusion and Decomposition Network) [6] tackles the challenge of subtle tampering in document images by jointly modeling spatial and frequency domains. It introduces a Visual Enhancement Module (VEM) that fuses visual features with frequency-aware representations

using an attention mechanism, and a Wavelet-like Frequency Enhancement (WFE) module that explicitly decomposes features into high- and low-frequency components to capture faint tampering traces. This dual-path architecture enhances both perceptibility and robustness. Evaluated on DocTamper and T-SROIE, FFDN significantly outperforms previous methods, especially in detecting small tampered regions under compression and noise.

### D.3 Metrics

**AP (Average Precision).** Average Precision (AP) is calculated as the area under the precision-recall curve. It is defined as:

$$AP = \int_0^1 \text{Precision}(r) \, dr$$

where $\text{Precision}(r)$ is the precision at recall level $r$. The precision is calculated as:

$$\text{Precision} = \frac{TP}{TP + FP}$$

and recall is:

$$\text{Recall} = \frac{TP}{TP + FN}$$

where $TP$ is true positives, $FP$ is false positives, and $FN$ is false negatives.

**MCC (Matthews Correlation Coefficient).** The Matthews Correlation Coefficient (MCC) is calculated as:

$$\text{MCC} = \frac{TP \cdot TN - FP \cdot FN}{\sqrt{(TP + FP)(TP + FN)(TN + FP)(TN + FN)}}$$

where $TP$ is true positives, $TN$ is true negatives, $FP$ is false positives, and $FN$ is false negatives.

**TNR (True Negative Rate).** The True Negative Rate (TNR) is defined as:

$$\text{TNR} = \frac{TN}{TN + FP}$$

where $TN$ is true negatives and $FP$ is false positives.

**TPR (True Positive Rate).** The True Positive Rate (TPR), also known as recall, is given by:

$$\text{TPR} = \frac{TP}{TP + FN}$$

where $TP$ is true positives and $FN$ is false negatives.

**AUC (Area Under the Curve).** The Area Under the Curve (AUC) is the area under the Receiver Operating Characteristic (ROC) curve. It can be calculated as:

$$\text{AUC} = \int_0^1 \text{TPR}(FPR) \, dFPR$$

where $\text{TPR}(FPR)$ is the true positive rate at a given false positive rate ($FPR$).

**ACC (Accuracy).** Accuracy is calculated as the ratio of correctly classified instances to the total number of instances:

$$\text{ACC} = \frac{TP + TN}{TP + TN + FP + FN}$$

where $TP$ is true positives, $TN$ is true negatives, $FP$ is false positives, and $FN$ is false negatives.

**F1 (F1 Score).** The F1 score is the harmonic mean of precision and recall, given by:

$$\text{F1} = 2 \cdot \frac{\text{Precision} \cdot \text{Recall}}{\text{Precision} + \text{Recall}}$$

where precision and recall are defined as:

$$\text{Precision} = \frac{TP}{TP + FP}, \quad \text{Recall} = \frac{TP}{TP + FN}$$

**IOU (Intersection over Union).** Intersection over Union (IoU) is calculated as the ratio of the intersection of predicted and ground truth regions to their union:

$$\text{IoU} = \frac{|A \cap B|}{|A \cup B|}$$

where $A$ is the predicted region and $B$ is the ground truth region.

# E    Details of AIGC and Document Benchmarks

## E.1    Training Details for AIGC Benchmark Implementation

We resize images to *224×224* and apply only basic data augmentations, including flipping, brightness and contrast adjustment, compression, and Gaussian blur. All models are trained for 20 epochs using a cosine decay learning rate schedule, decreasing from $1e-4$ to $1e-5$. We use the ImageNet split of DiffusionForensics as the training set. For Synthbuster, we use a fully connected layer as the classifier.

## E.2    Training Details for Document Image Manipulation Localization Benchmark Models

For all models, we adopt a cosine learning rate schedule decaying from $1e-4$ to $5e-7$, using the AdamW optimizer with $\beta_1$=0.9, $\beta_2$=0.999, weight decay of 0.05, and gradient accumulation step of 1.

Epoch schedules are adapted to dataset size and complexity: 10 epochs for Doctamper, 75 epochs for RTM, and 150 epochs for all other datasets. We use a batch size of 10 for CATFB, 8 for DTD and FFDN, and 4 for TIFDM.

Table 11: Cross-dataset AUC evaluation on Deepfake benchmarks.

| Model | FF++c40 | FF-DF | FF-F2F | FF-NT | FF-FS | CDFv1 | CDFv2 | DFD | DFDC | DFDCP | FaceShifter | UADFV | Average |
|---|---|---|---|---|---|---|---|---|---|---|---|---|---|
| ResNet | 0.675 | 0.938 | 0.936 | 0.879 | 0.935 | 0.692 | 0.684 | 0.834 | 0.678 | 0.763 | 0.613 | 0.770 | 0.783 |
| Xception | 0.728 | 0.976 | 0.967 | 0.925 | 0.971 | 0.708 | 0.719 | 0.870 | 0.702 | 0.764 | 0.605 | 0.885 | 0.818 |
| EfficientNet | 0.504 | 0.503 | 0.505 | 0.508 | 0.500 | 0.541 | 0.535 | 0.517 | 0.510 | 0.454 | 0.478 | 0.480 | 0.503 |
| Swin-Transformer | 0.771 | 0.992 | 0.988 | 0.969 | 0.988 | 0.704 | 0.746 | 0.901 | 0.754 | 0.820 | 0.655 | 0.721 | 0.834 |
| SegFormer | 0.739 | 0.993 | 0.990 | 0.962 | 0.993 | 0.799 | 0.799 | 0.885 | 0.726 | 0.771 | 0.719 | 0.889 | 0.853 |
| ConvNeXt | 0.794 | 0.996 | 0.991 | 0.981 | 0.994 | 0.718 | 0.784 | 0.911 | 0.747 | 0.842 | 0.756 | 0.805 | 0.860 |
| CapsuleNet | 0.613 | 0.855 | 0.867 | 0.839 | 0.800 | 0.527 | 0.660 | 0.699 | 0.640 | 0.656 | 0.624 | 0.761 | 0.712 |
| Recce | 0.634 | 0.815 | 0.843 | 0.762 | 0.848 | 0.572 | 0.602 | 0.727 | 0.625 | 0.672 | 0.531 | 0.816 | 0.704 |
| Spsl | 0.730 | 0.982 | 0.966 | 0.933 | 0.974 | 0.690 | 0.726 | 0.876 | 0.706 | 0.758 | 0.632 | 0.878 | 0.821 |
| MVSS-Net | 0.713 | 0.992 | 0.981 | 0.953 | 0.986 | 0.655 | 0.700 | 0.857 | 0.707 | 0.727 | 0.696 | 0.737 | 0.809 |
| Trufor | 0.642 | 0.968 | 0.967 | 0.931 | 0.960 | 0.683 | 0.698 | 0.832 | 0.670 | 0.769 | 0.629 | 0.746 | 0.791 |
| IML-ViT | 0.750 | 0.973 | 0.963 | 0.906 | 0.973 | 0.763 | 0.726 | 0.851 | 0.708 | 0.663 | 0.560 | 0.702 | 0.795 |
| Mesorch | 0.767 | 0.991 | 0.983 | 0.956 | 0.983 | 0.803 | 0.814 | 0.867 | 0.760 | 0.807 | 0.665 | 0.889 | 0.857 |
| DualNet | 0.637 | 0.784 | 0.685 | 0.707 | 0.606 | 0.471 | 0.552 | 0.540 | 0.547 | 0.559 | 0.667 | 0.530 | 0.607 |
| HifiNet | 0.587 | 0.727 | 0.664 | 0.646 | 0.598 | 0.456 | 0.611 | 0.648 | 0.607 | 0.619 | 0.544 | 0.745 | 0.621 |
| UnivFD | 0.690 | 0.931 | 0.645 | 0.549 | 0.871 | 0.660 | 0.671 | 0.798 | 0.633 | 0.674 | 0.656 | 0.891 | 0.722 |
| DTD | 0.498 | 0.509 | 0.522 | 0.513 | 0.507 | 0.576 | 0.520 | 0.490 | 0.485 | 0.499 | 0.484 | 0.499 | 0.508 |
| FFDN | 0.714 | 0.997 | 0.991 | 0.980 | 0.997 | 0.706 | 0.699 | 0.871 | 0.712 | 0.769 | 0.741 | 0.771 | 0.829 |

# F    Details of IFF-Protocol

**Implementation Resolution.**    We use the commonly used *256×256* resolution for detection tasks, such as Deepfake and AIGC. However, UnivFD uses CLIP-ViT as the backbone, which only supports *224×224* image input. Therefore, the input image is resized to *224×224* for UnivFD. On the other hand, the SoTA for Document is specifically designed for *512×512* resolution, with some models like FFDN even having a fixed input size of *512×512*. Therefore, we resize images to *512×512* for Document models.

**Results on Domains.**    We provide the test results of backbones and domain-specific SoTAs under the IFF-Protocol for each domain, which are Table 11 for Deepfake, Table 12 for IMDL, Table 13 for AIGC, and Table 14 for Document.

**Experiments on Recent Datasets.**    We added experiments on the DF40 [74] and Chameleon [71] dataset, along with evaluations of two recent Deepfake SoTAs: Sia [60] (ECCV22) and Effort [73] (ICML25), and two recent AIGC SoTAs: FatFormer [33] (CVPR24) and CO-SPY [7] (CVPR25). Results are shown in Table 15.

Table 12: Cross-dataset AUC evaluation on Image Manipulation Detection and Localization (IMDL) datasets.

| Model | COVERAGE | Columbia | IMD2020 | NIST16 | Cocoglide | Autosplice | Average |
|---|---|---|---|---|---|---|---|
| ResNet | 0.492 | 0.380 | 0.549 | 0.540 | 0.629 | 0.764 | 0.559 |
| Xception | 0.491 | 0.465 | 0.537 | 0.582 | 0.666 | 0.756 | 0.583 |
| EfficientNet | 0.489 | 0.623 | 0.506 | 0.496 | 0.508 | 0.483 | 0.517 |
| Swin-Transformer | 0.505 | 0.636 | 0.631 | 0.361 | 0.816 | 0.864 | 0.635 |
| SegFormer | 0.501 | 0.558 | 0.594 | 0.502 | 0.785 | 0.819 | 0.627 |
| ConvNeXt | 0.489 | 0.625 | 0.598 | 0.529 | 0.761 | 0.825 | 0.638 |
| CapsuleNet | 0.504 | 0.330 | 0.527 | 0.486 | 0.704 | 0.745 | 0.549 |
| Recce | 0.485 | 0.506 | 0.492 | 0.593 | 0.616 | 0.642 | 0.556 |
| Spsl | 0.493 | 0.419 | 0.545 | 0.557 | 0.675 | 0.759 | 0.575 |
| MVSS-Net | 0.481 | 0.298 | 0.539 | 0.527 | 0.683 | 0.795 | 0.554 |
| Trufor | 0.492 | 0.306 | 0.564 | 0.368 | 0.710 | 0.808 | 0.541 |
| IML-ViT | 0.498 | 0.483 | 0.556 | 0.484 | 0.657 | 0.819 | 0.583 |
| Mesorch | 0.495 | 0.285 | 0.570 | 0.494 | 0.653 | 0.773 | 0.545 |
| DualNet | 0.503 | 0.268 | 0.517 | 0.314 | 0.619 | 0.748 | 0.495 |
| HifiNet | 0.500 | 0.745 | 0.534 | 0.503 | 0.589 | 0.677 | 0.591 |
| UnivFD | 0.499 | 0.886 | 0.786 | 0.715 | 0.706 | 0.785 | 0.729 |
| DTD | 0.500 | 0.679 | 0.498 | 0.468 | 0.510 | 0.506 | 0.527 |
| FFDN | 0.522 | 0.553 | 0.624 | 0.574 | 0.681 | 0.927 | 0.647 |

Table 13: Cross-domain AUC evaluation on AIGC datasets.

| Model | DiffusionForensics | GenImage_all | GenImage_ADM | GenImage_Midjourney | GenImage_wukong | GenImage_glide | GenImage_BigGAN | GenImage_sd14 | GenImage_sd15 | GenImage_VQDM | Average |
|---|---|---|---|---|---|---|---|---|---|---|---|
| ResNet | 0.614 | 0.865 | 0.649 | 0.818 | 0.928 | 0.936 | 0.971 | 0.919 | 0.920 | 0.761 | 0.838 |
| Xception | 0.757 | 0.980 | 0.990 | 0.949 | 0.979 | 0.986 | 0.997 | 0.987 | 0.985 | 0.963 | 0.957 |
| EfficientNet | 0.544 | 0.597 | 0.842 | 0.485 | 0.642 | 0.523 | 0.542 | 0.619 | 0.619 | 0.489 | 0.590 |
| Swin-Transformer | 0.915 | 0.999 | 1.000 | 0.998 | 0.999 | 1.000 | 1.000 | 1.000 | 1.000 | 1.000 | 0.991 |
| SegFormer | 0.847 | 0.998 | 1.000 | 0.995 | 0.997 | 0.999 | 1.000 | 0.999 | 0.999 | 0.999 | 0.983 |
| ConvNeXt | 0.895 | 1.000 | 1.000 | 0.999 | 1.000 | 1.000 | 1.000 | 1.000 | 1.000 | 1.000 | 0.989 |
| CapsuleNet | 0.546 | 0.971 | 0.992 | 0.921 | 0.971 | 0.978 | 0.992 | 0.984 | 0.982 | 0.944 | 0.928 |
| Recce | 0.684 | 0.906 | 0.909 | 0.856 | 0.927 | 0.907 | 0.948 | 0.933 | 0.936 | 0.819 | 0.882 |
| Spsl | 0.770 | 0.987 | 0.994 | 0.965 | 0.985 | 0.990 | 0.999 | 0.992 | 0.991 | 0.979 | 0.965 |
| MVSS-Net | 0.671 | 0.994 | 0.999 | 0.978 | 0.991 | 0.996 | 1.000 | 0.996 | 0.996 | 0.996 | 0.962 |
| Trufor | 0.726 | 0.996 | 0.998 | 0.990 | 0.992 | 0.997 | 0.998 | 0.997 | 0.997 | 0.997 | 0.969 |
| IML-ViT | 0.627 | 0.991 | 0.999 | 0.960 | 0.988 | 0.989 | 0.999 | 0.996 | 0.997 | 0.997 | 0.954 |
| Mesorch | 0.629 | 0.996 | 0.999 | 0.987 | 0.995 | 0.998 | 0.999 | 0.998 | 0.997 | 0.997 | 0.960 |
| DualNet | 0.899 | 0.988 | 0.999 | 0.926 | 0.992 | 0.993 | 0.999 | 0.997 | 0.996 | 0.998 | 0.979 |
| HifiNet | 0.575 | 0.756 | 0.707 | 0.618 | 0.867 | 0.830 | 0.751 | 0.820 | 0.812 | 0.618 | 0.735 |
| UnivFD | 0.742 | 0.813 | 0.767 | 0.725 | 0.829 | 0.852 | 0.883 | 0.841 | 0.839 | 0.759 | 0.805 |
| DTD | 0.457 | 0.499 | 0.542 | 0.547 | 0.510 | 0.531 | 0.459 | 0.518 | 0.505 | 0.379 | 0.495 |
| FFDN | 0.999 | 1.000 | 1.000 | 1.000 | 1.000 | 1.000 | 1.000 | 1.000 | 1.000 | 1.000 | 1.000 |

**Common features and conflicting patterns across domains.** We selected two IMDL models: MVSS-Net and IML-ViT, and used IFF-Protocol weights (where models are trained across-domain) as pretrained weights. These models were then trained on the IMDL task to investigate whether the artifacts learned across domains could benefit finetuning within a single domain. Results are shown in the Table 16.

# G   Details of Experiments

## G.1   Details of Feature Extractors

**BayarConv.**   BayarConv [3] is a constrained convolutional layer, that is able to jointly suppress an image's content and adaptively learn manipulation detection features. It learns to extract noise artifacts within images.

**Sobel.**   Sobel layer is proposed to enhance edge-related patterns, whereas the subtle boundary cues are critical for manipulation detection and localization [5]. This is based on the common assumption that manipulations often leave edge artifacts along the tampered boundaries.

**DCT.**   DCT (Discrete Cosine Transform) [1] is a mathematical technique that transforms spatial domain data into frequency domain components, primarily used to isolate image features based on their frequency to extract frequency features.

**FPH.**   FPH [48] is designed to find out tampering clues in the frequency domain with DCT coefficients. It receives DCT coefficients and a quantization table as input, and outputs a 256-channel feature map that is downsampled by a factor of 8. This design enables it to effectively capture compression artifacts and frequency-domain inconsistencies for downstream analysis.

Table 14: Cross-dataset AUC evaluation on document manipulation datasets.

| Model | OSTF | RTM | T-SROIE | Tampered-IC13 | Average |
|---|---|---|---|---|---|
| ResNet | 0.704 | 0.689 | 0.953 | 0.807 | 0.788 |
| Xception | 0.762 | 0.734 | 0.966 | 0.877 | 0.835 |
| EfficientNet | 0.581 | 0.512 | 0.884 | 0.653 | 0.657 |
| Swin-Transformer | 0.856 | 0.758 | 0.990 | 0.966 | 0.893 |
| SegFormer | 0.856 | 0.705 | 0.980 | 0.956 | 0.874 |
| ConvNeXt | 0.849 | 0.762 | 0.994 | 0.957 | 0.890 |
| CapsuleNet | 0.704 | 0.670 | 0.946 | 0.759 | 0.770 |
| Recce | 0.688 | 0.555 | 0.542 | 0.797 | 0.645 |
| Spsl | 0.769 | 0.738 | 0.972 | 0.888 | 0.842 |
| MVSS-Net | 0.806 | 0.741 | 0.978 | 0.913 | 0.860 |
| Trufor | 0.805 | 0.732 | 0.979 | 0.920 | 0.859 |
| IML-ViT | 0.800 | 0.703 | 0.972 | 0.923 | 0.850 |
| Mesorch | 0.819 | 0.739 | 0.982 | 0.954 | 0.873 |
| DualNet | 0.658 | 0.657 | 0.935 | 0.776 | 0.756 |
| HifiNet | 0.663 | 0.615 | 0.937 | 0.762 | 0.744 |
| UnivFD | 0.684 | 0.569 | 0.938 | 0.740 | 0.733 |
| DTD | 0.595 | 0.496 | 0.748 | 0.658 | 0.624 |
| FFDN | 0.893 | 0.782 | 0.997 | 0.975 | 0.912 |

Table 15: AUC performance on recent DF40 and Chameleon datasets.

| Model | DF40_CollabDiff | DF40_deepfacelab | DF40_heygen | Chameleon |
|---|---|---|---|---|
| ConvNeXt | 0.935 | 0.795 | 0.744 | 0.626 |
| Capsule-Net | 0.984 | 0.638 | 0.714 | 0.676 |
| MVSS-Net | 0.983 | 0.699 | 0.456 | 0.725 |
| UnivFD | 0.629 | 0.752 | 0.946 | 0.798 |
| DTD | 0.494 | 0.477 | 0.517 | 0.631 |
| Sia | 0.808 | 0.764 | 0.641 | 0.569 |
| Effort | 0.995 | 0.894 | 0.949 | 0.604 |
| FatFormer | 0.983 | 0.716 | 0.611 | 0.707 |
| CO-SPY | 0.890 | 0.792 | 0.656 | 0.767 |

### G.2 Details for Extractor & Backbone in different tasks

We provide the performance differences of 6 backbones with and without 4 different feature extractors across the 4 domains. The table presents detailed results for each individual test dataset. They are Table 17 for IMDL, Table 18 for AIGC, Tabel 19 for Deepfake, and Table 20 for Document.

## H Computational Resources

The experiments were conducted on three different servers. The first server is equipped with two AMD EPYC 7542 CPUs, 256GB RAM, and 6×NVIDIA A40 GPUs, which was used for all IFF-related experiments. The remaining experiments were performed on two servers: one with a single AMD EPYC 7542 CPU, 256GB RAM, and 4×NVIDIA RTX 3090 GPUs, and another with two AMD EPYC 7542 CPUs, 256GB RAM, and 8×NVIDIA RTX 3090 GPUs.

## I Broader Impacts Discussion

ForensicHub establishes a critical benchmark for all-domain fake image detection and localization, helping to curb the spread of falsified images in society and significantly advancing the development of a more trustworthy digital environment. However, the comprehensive coverage of detection methods in ForensicHub may enable malicious actors to study and develop targeted evasion techniques.

Table 16: Common features and conflicting patterns across domains.

| Method | Coverage | Columbia | NIST16 | CASIAv1 | IMD2020 | Avg. |
|---|---|---|---|---|---|---|
| MVSS-Net (Image-Net) | 0.259 | 0.386 | 0.246 | 0.534 | 0.279 | 0.341 |
| MVSS-Net (IFF-Protocol) | 0.268 | 0.395 | 0.259 | 0.562 | 0.292 | 0.355 |
| IML-ViT (Image-Net) | 0.435 | 0.780 | 0.331 | 0.721 | 0.327 | 0.519 |
| IML-ViT (IFF-Protocol) | 0.427 | 0.767 | 0.279 | 0.715 | 0.351 | 0.508 |

Table 17: Extractor & backbone performance difference in IMDL region

| Extractor | Model | Coverage | Columbia | IMD2020 | NIST16 | Cocoglide | Autosplice | Mean Diff |
|---|---|---|---|---|---|---|---|---|
| Sobel [5] | ResNet | 0.018 | 0.141 | -0.046 | 0.084 | -0.048 | -0.093 | 0.009 |
| | Xception | -0.005 | -0.001 | 0.015 | -0.010 | -0.052 | -0.069 | -0.020 |
| | EfficientNet | 0.011 | -0.045 | -0.018 | 0.060 | 0.015 | 0.012 | 0.006 |
| | Swin | -0.006 | -0.192 | -0.118 | 0.256 | -0.196 | -0.247 | -0.084 |
| | SegFormer | -0.018 | -0.007 | -0.071 | 0.091 | -0.191 | -0.189 | -0.064 |
| | ConvNeXt | 0.012 | -0.019 | -0.101 | 0.023 | -0.146 | -0.181 | -0.069 |
| Bayar [3] | ResNet | -0.007 | 0.027 | -0.025 | 0.064 | -0.044 | -0.086 | -0.012 |
| | Xception | -0.009 | -0.049 | -0.001 | -0.026 | -0.007 | -0.044 | -0.023 |
| | EfficientNet | 0.019 | 0.004 | 0.007 | 0.006 | 0.002 | 0.013 | 0.009 |
| | Swin | 0.004 | -0.282 | -0.128 | 0.214 | -0.156 | -0.215 | -0.094 |
| | SegFormer | 0.009 | -0.108 | -0.069 | 0.073 | -0.173 | -0.173 | -0.073 |
| | ConvNeXt | 0.005 | -0.234 | -0.080 | 0.028 | -0.126 | -0.141 | -0.091 |
| FPH [48] | ResNet | -0.002 | 0.169 | -0.044 | -0.016 | -0.079 | -0.105 | -0.013 |
| | Xception | 0.007 | -0.016 | -0.011 | 0.013 | -0.102 | -0.035 | -0.024 |
| | EfficientNet | 0.017 | -0.062 | -0.002 | 0.095 | 0.057 | 0.115 | 0.037 |
| | Swin | -0.010 | -0.208 | -0.115 | 0.235 | -0.248 | -0.204 | -0.092 |
| | SegFormer | -0.007 | -0.104 | -0.066 | 0.077 | -0.263 | -0.197 | -0.093 |
| | ConvNeXt | 0.014 | 0.203 | -0.097 | -0.044 | -0.272 | -0.313 | -0.085 |
| DCT [1] | ResNet | -0.002 | 0.073 | -0.005 | 0.023 | 0.002 | -0.030 | 0.010 |
| | Xception | 0.015 | 0.025 | -0.002 | 0.001 | 0.009 | -0.008 | 0.007 |
| | EfficientNet | 0.007 | 0.034 | -0.007 | 0.018 | -0.005 | 0.005 | 0.009 |
| | Swin | -0.023 | -0.162 | -0.066 | 0.206 | -0.189 | -0.105 | -0.057 |
| | SegFormer | -0.003 | -0.156 | -0.021 | 0.067 | -0.140 | -0.043 | -0.049 |
| | ConvNeXt | -0.002 | -0.187 | -0.078 | 0.038 | -0.216 | -0.145 | -0.098 |

Table 18: Extractor & backbone performance difference in AIGC region

| extractor | model | DiffusionForensics_test | GenImage_all | GenImage_ADM | GenImage_Midjourney | GenImage_wukong | GenImage_glide | GenImage_BigGAN | GenImage_sd14 | GenImage_sd15 | GenImage_VQDM | mean_diff |
|---|---|---|---|---|---|---|---|---|---|---|---|---|
| Sobel [5] | ResNet | -0.037 | -0.046 | 0.312 | -0.112 | -0.064 | -0.073 | -0.186 | -0.060 | -0.061 | -0.119 | -0.045 |
| | Xception | -0.133 | -0.042 | -0.007 | -0.090 | -0.031 | -0.020 | -0.007 | -0.035 | -0.033 | -0.115 | -0.051 |
| | EfficientNet | 0.049 | 0.031 | 0.015 | 0.029 | -0.023 | 0.128 | 0.105 | -0.023 | -0.036 | 0.084 | 0.036 |
| | Swin | -0.206 | -0.086 | -0.014 | -0.161 | -0.073 | -0.035 | -0.036 | -0.081 | -0.079 | -0.212 | -0.098 |
| | SegFormer | -0.181 | -0.095 | -0.025 | -0.185 | -0.085 | -0.055 | -0.056 | -0.083 | -0.079 | -0.203 | -0.105 |
| | ConvNeXt | -0.212 | -0.129 | -0.027 | -0.278 | -0.117 | -0.069 | -0.004 | -0.124 | -0.124 | -0.290 | -0.137 |
| Bayar [3] | ResNet | -0.027 | -0.060 | 0.007 | -0.054 | -0.050 | -0.029 | -0.165 | -0.042 | -0.039 | -0.123 | -0.058 |
| | Xception | -0.054 | -0.023 | -0.018 | -0.034 | -0.019 | -0.009 | -0.006 | -0.019 | -0.019 | -0.057 | -0.026 |
| | EfficientNet | 0.013 | -0.013 | -0.096 | 0.008 | -0.001 | 0.001 | -0.033 | 0.003 | 0.004 | 0.009 | -0.011 |
| | Swin | -0.240 | -0.052 | -0.008 | -0.117 | -0.058 | -0.038 | -0.004 | -0.054 | -0.056 | -0.082 | -0.071 |
| | SegFormer | -0.215 | -0.054 | -0.005 | -0.113 | -0.057 | -0.043 | -0.010 | -0.049 | -0.050 | -0.109 | -0.070 |
| | ConvNeXt | -0.227 | -0.050 | -0.015 | -0.247 | -0.055 | -0.036 | -0.007 | -0.051 | -0.053 | -0.055 | -0.067 |
| FPH [48] | ResNet | -0.108 | -0.132 | -0.094 | -0.186 | -0.060 | -0.130 | -0.195 | -0.082 | -0.094 | -0.172 | -0.131 |
| | Xception | -0.198 | -0.094 | -0.033 | -0.044 | -0.084 | -0.097 | -0.049 | -0.048 | -0.227 | -0.106 |
| | EfficientNet | 0.031 | 0.153 | 0.051 | 0.118 | 0.163 | 0.319 | 0.315 | 0.124 | 0.124 | 0.027 | 0.142 |
| | Swin | -0.264 | -0.073 | -0.006 | -0.153 | -0.035 | -0.073 | -0.073 | -0.035 | -0.036 | -0.192 | -0.094 |
| | SegFormer | -0.217 | -0.087 | -0.018 | -0.201 | -0.048 | -0.088 | -0.081 | -0.045 | -0.046 | -0.187 | -0.102 |
| | ConvNeXt | -0.452 | -0.511 | -0.547 | -0.507 | -0.399 | -0.539 | -0.581 | -0.452 | -0.456 | -0.627 | -0.507 |
| DCT [1] | ResNet | -0.036 | -0.015 | 0.011 | -0.016 | -0.002 | -0.020 | -0.036 | -0.010 | -0.014 | -0.035 | -0.017 |
| | Xception | -0.007 | 0.000 | 0.000 | 0.000 | 0.003 | 0.000 | -0.001 | 0.000 | 0.000 | -0.004 | -0.001 |
| | EfficientNet | 0.006 | 0.008 | -0.017 | 0.009 | 0.013 | 0.004 | -0.009 | 0.021 | 0.016 | 0.030 | 0.008 |
| | Swin | -0.253 | -0.019 | -0.002 | -0.042 | -0.013 | -0.019 | -0.011 | -0.008 | -0.011 | -0.051 | -0.043 |
| | SegFormer | -0.223 | -0.012 | -0.004 | -0.029 | -0.014 | -0.011 | -0.006 | -0.007 | -0.008 | -0.025 | -0.034 |
| | ConvNeXt | -0.234 | -0.051 | -0.012 | -0.109 | -0.031 | -0.062 | -0.045 | -0.018 | -0.019 | -0.123 | -0.070 |

Table 19: Extractor & backbone performance difference in Deepfake region

| Extractor | Model | FF++c40 | FF-DF | FF-F2F | FF-NT | FF-FS | CDFv1 | CDFv2 | DFD | DFDC | DFDCP | FaceShifter | UADFV | Mean |
|---|---|---|---|---|---|---|---|---|---|---|---|---|---|---|
| Sobel [5] | ResNet [21] | -0.052 | -0.111 | -0.166 | -0.150 | -0.212 | -0.097 | -0.042 | -0.144 | -0.076 | -0.063 | -0.046 | 0.095 | -0.089 |
| | Xception [8] | 0.022 | -0.021 | -0.034 | -0.056 | -0.040 | 0.033 | 0.034 | -0.078 | -0.009 | -0.023 | -0.004 | -0.020 | -0.016 |
| | EfficientNet [61] | 0.022 | 0.044 | 0.034 | 0.035 | 0.017 | -0.031 | -0.017 | -0.006 | 0.043 | 0.050 | 0.069 | 0.068 | 0.027 |
| | Swin-Transformer [35] | -0.130 | -0.133 | -0.197 | -0.232 | -0.255 | 0.002 | -0.084 | -0.234 | -0.102 | -0.142 | 0.001 | -0.077 | -0.132 |
| | SegFormer [70] | -0.040 | -0.117 | -0.174 | -0.224 | -0.180 | -0.034 | -0.145 | -0.176 | -0.089 | -0.111 | -0.104 | -0.120 | -0.126 |
| | ConvNeXt [36] | -0.141 | -0.120 | -0.169 | -0.222 | -0.232 | -0.066 | -0.167 | -0.228 | -0.115 | -0.198 | -0.095 | -0.235 | -0.166 |
| Bayar [3] | ResNet [21] | -0.006 | -0.071 | -0.081 | -0.097 | -0.175 | -0.100 | -0.042 | -0.122 | -0.044 | -0.088 | -0.119 | 0.046 | -0.075 |
| | Xception [8] | 0.006 | -0.007 | -0.013 | -0.021 | -0.011 | 0.006 | -0.007 | -0.023 | -0.015 | -0.025 | -0.008 | -0.005 | -0.010 |
| | EfficientNet [61] | 0.000 | -0.011 | 0.001 | 0.000 | 0.002 | 0.064 | 0.030 | -0.003 | 0.024 | -0.018 | 0.012 | 0.004 | 0.009 |
| | Swin-Transformer [35] | -0.050 | -0.092 | -0.134 | -0.185 | -0.213 | 0.056 | -0.032 | -0.107 | -0.097 | -0.168 | -0.128 | -0.006 | -0.096 |
| | SegFormer [70] | 0.023 | -0.069 | -0.119 | -0.154 | -0.109 | -0.040 | -0.123 | -0.077 | -0.077 | -0.084 | -0.177 | -0.162 | -0.097 |
| | ConvNeXt [36] | -0.019 | -0.062 | -0.109 | -0.151 | -0.126 | 0.028 | -0.052 | -0.113 | -0.085 | -0.103 | -0.200 | -0.061 | -0.088 |
| FPH [48] | ResNet [21] | -0.019 | -0.123 | -0.305 | -0.268 | -0.367 | -0.105 | -0.108 | -0.214 | -0.094 | -0.074 | 0.013 | 0.008 | -0.138 |
| | Xception [8] | -0.010 | -0.106 | -0.173 | -0.137 | -0.137 | 0.094 | -0.052 | -0.211 | -0.065 | -0.206 | -0.141 | -0.085 | -0.100 |
| | EfficientNet [61] | 0.054 | 0.123 | 0.031 | 0.052 | 0.017 | -0.057 | 0.044 | 0.078 | 0.045 | 0.143 | 0.104 | 0.081 | 0.060 |
| | Swin-Transformer [35] | 0.037 | -0.066 | -0.117 | -0.252 | -0.093 | 0.078 | -0.008 | -0.161 | -0.072 | -0.068 | -0.104 | 0.124 | -0.059 |
| | SegFormer [70] | 0.062 | -0.108 | -0.201 | -0.330 | -0.164 | -0.067 | -0.153 | -0.271 | -0.089 | -0.159 | -0.111 | -0.094 | -0.140 |
| | ConvNeXt | -0.282 | -0.482 | -0.498 | -0.468 | -0.473 | -0.150 | -0.368 | -0.380 | -0.225 | -0.296 | -0.237 | -0.272 | -0.344 |
| DCT [1] | ResNet [21] | 0.014 | -0.006 | -0.008 | -0.013 | -0.011 | 0.052 | 0.029 | -0.029 | -0.001 | 0.011 | 0.007 | 0.021 | 0.005 |
| | Xception [8] | -0.006 | 0.004 | -0.003 | 0.004 | -0.001 | -0.020 | 0.006 | -0.011 | 0.005 | -0.023 | 0.003 | -0.032 | -0.006 |
| | EfficientNet [61] | -0.004 | -0.004 | 0.005 | 0.004 | 0.008 | 0.035 | 0.023 | -0.006 | 0.036 | 0.011 | 0.020 | 0.001 | 0.011 |
| | Swin-Transformer [35] | 0.040 | -0.027 | -0.030 | -0.105 | -0.027 | 0.092 | -0.021 | -0.060 | 0.001 | -0.045 | -0.071 | 0.159 | -0.008 |
| | SegFormer [70] | 0.043 | -0.019 | -0.028 | -0.041 | -0.026 | 0.039 | -0.037 | -0.039 | 0.007 | 0.007 | -0.116 | 0.026 | -0.015 |
| | ConvNeXt [36] | 0.041 | -0.049 | -0.087 | -0.173 | -0.076 | -0.102 | -0.127 | -0.158 | -0.025 | -0.135 | -0.288 | -0.013 | -0.099 |

Table 20: Extractor & backbone performance difference in Document region

| extractor | model | OSTF_test | RealTextManipulation_test | T-SROIE_test | Tampered-IC13_test | mean_diff |
|---|---|---|---|---|---|---|
| Sobel [5] | ResNet | -0.044 | -0.046 | -0.020 | -0.070 | -0.045 |
| | Xception | -0.046 | -0.019 | -0.012 | -0.030 | -0.027 |
| | EfficientNet | -0.025 | 0.033 | 0.019 | -0.002 | 0.006 |
| | Swin | -0.143 | -0.051 | -0.033 | -0.153 | -0.095 |
| | SegFormer | -0.135 | -0.008 | -0.027 | -0.156 | -0.082 |
| | ConvNeXt | -0.160 | -0.090 | -0.041 | -0.200 | -0.123 |
| Bayar [3] | ResNet | -0.023 | -0.050 | -0.013 | -0.047 | -0.033 |
| | Xception | 0.001 | -0.015 | -0.010 | 0.001 | -0.006 |
| | EfficientNet | -0.001 | 0.003 | 0.007 | 0.001 | 0.003 |
| | Swin | -0.119 | -0.040 | -0.040 | -0.105 | -0.076 |
| | SegFormer | -0.123 | 0.002 | -0.033 | -0.107 | -0.065 |
| | ConvNeXt | -0.145 | -0.047 | -0.036 | -0.126 | -0.088 |
| FPH [48] | ResNet | -0.053 | -0.071 | -0.009 | -0.077 | -0.052 |
| | Xception | -0.079 | -0.045 | -0.012 | -0.122 | -0.065 |
| | EfficientNet | 0.049 | 0.023 | -0.053 | 0.044 | 0.016 |
| | Swin | -0.113 | -0.034 | -0.038 | -0.091 | -0.069 |
| | SegFormer | -0.202 | 0.014 | -0.022 | -0.176 | -0.097 |
| | ConvNeXt | -0.236 | -0.209 | -0.072 | -0.289 | -0.202 |
| DCT [1] | ResNet | -0.008 | -0.005 | 0.004 | -0.026 | -0.009 |
| | Xception | -0.001 | 0.001 | -0.002 | 0.002 | 0.000 |
| | EfficientNet | -0.013 | 0.028 | 0.007 | 0.003 | 0.006 |
| | Swin | -0.093 | -0.028 | -0.020 | -0.056 | -0.049 |
| | SegFormer | -0.063 | 0.018 | -0.011 | -0.029 | -0.021 |
| | ConvNeXt | -0.132 | -0.050 | -0.039 | -0.107 | -0.082 |

