# OpenReview forum: "ForensicHub: A Unified Benchmark & Codebase for All-Domain Fake Image Detection and Localization"
_NeurIPS.cc/2025/Datasets_and_Benchmarks_Track — NeurIPS 2025 Datasets and Benchmarks Track poster_

### Official Review · Reviewer_4Hjf · 2025-06-23

**Rating:** 4
**Confidence:** 4

**Summary:**

This paper presents ForensicHub, a unified benchmark and codebase encompassing four key areas—deepfake detection, image manipulation detection and localization, AI-generated content detection, and document image manipulation, and provides a systematic analysis of existing forensic detectors across datasets for these tasks.

**Additional Feedback:**

There are some typos, for example “Faceforencs++” on line 129.

**Dataset Code Accessibility:**

Yes

**Dataset Code Comments:**

The codes are accessible.

**Ethical Considerations:**

No, there are no or only very minor ethics concerns

**Final Justification:**

I have raised my score since the authors have addressed most of my concerns and satisfactorily responded to the other reviewers' comments.

**Limitations Weaknesses:**

One major issue with ForensicHub is that, although it consolidates deepfake detection, AIGC detection, image manipulation detection and localization (IMDL), and document manipulation localization under a single evaluation framework and performs cross-task benchmarking, these four tasks exhibit fundamentally different characteristics. Evaluating them side-by-side in a uniform pipeline offers limited insight unless the paper also proposes a truly unified forensic framework—e.g., a multi-task architecture or an integral forensic system capable of handling all modalities. Some approaches that leverage MLLM have already achieved this. As it stands, simply aggregating disparate tasks into one benchmark is less impactful than delivering a deep, task-specific contribution that excels in a single domain.

While ForensicHub makes a commendable effort towards unifying the evaluation of fake image detection, a concern arises regarding the potentially overly uniform preprocessing strategy. For instance, the GenImage dataset, as discussed in 'Fake or JPEG? Revealing Common Biases in Generated Image Detection Datasets,' benefits from JPEG compression to mitigate dataset biases. Conversely, 'Improving Synthetic Image Detection Towards Generalization: An Image Transformation Perspective' suggests that JPEG compression can interfere with the artifacts forensic detectors aim to learn. The current approach of applying a standardized set of basic augmentations, while facilitating direct comparison, does not adequately reflect the diverse preprocessing requirements or sensitivities of different forensic detectors. This could inadvertently influence the learned artifacts and, consequently, the observed generalization capabilities. Maintaining flexibility in preprocessing, tailored to the characteristics of different detectors, could be crucial for a more nuanced understanding of their individual strengths and weaknesses.

The AIGC detectors used are outdated and lack comparisons with recently open-sourced state-of-the-art models. Additionally, although DeepfakeBench includes more recent deepfake detection methods, these have not been incorporated into the cross-task evaluations.

**Strengths Contributions:**

This work addresses a gap in fake image detection research by breaking down the significant domain silos where each of the four forensic domains independently develops datasets, models, and evaluation protocols. The paper demonstrates well organisation with experimental coverage and effective visual presentation through color-coded tables that clearly distinguish different forensic domains. ForensicHub demonstrates good extensibility through its modular architecture that decomposes forensic pipelines into interchangeable components, allowing flexible composition across datasets, models, and evaluators.

---

> ### Author Rebuttal · Authors · 2025-07-29
>
> Thank you for your valuable feedback and thoughtful comments. We sincerely appreciate your **recognition of our paper's structure and code implementation**, as well as your acknowledgment of our work's **role in breaking down critical domain silos in image forensics**. Here is our response to address your concerns.
>
> **Q1: Four tasks exhibit fundamentally different characteristics, and a unified framework offers limited insight.**
>
> A1: **Your concerns about our work actually come from the misunderstanding of the "artifacts" in image forensics.** We have accordingly clarified it in the below.
>
> **For all sub-domains within FIDL, models are designed for identifying the artifacts left by faking operations**, which is consistent across models in all domains [1-6]. To capture artifacts, models across domains often share similar pre-trained backbones and low-level visual feature extractors (as discussed in line 46 of the paper). *This view is also supported by other reviewers ('different yet similar forensic tasks' and 'long-standing issue of fragmentation across four sub-domains').*
>
> From this perspective, a unified benchmark can provide insights into generalized artifacts, thereby promoting progress across the entire FIDL field. This perspective is also acknowledged by other reviewers: *'The unified framework allows the community to examine whether detectors trained on domain-specific tasks—such as deepfake detection—can generalize to other types of forensic tasks. This offers potential insights into transferability.'*
>
> Besides, in real-world scenarios, it is often impossible **to predetermine the type of manipulation** (deepfake, imdl, aigc and document) present in an image, **making unified detection particularly important for users**.
>
> **Q2: Some approaches that leverage MLLM have already achieved unified forensic.**
>
> A2: To the best of our knowledge, **we have not yet seen any MLLM capable of unified forensics** that can simultaneously handle deepfake, IMDL, AIGC, and document domains. Existing MLLM approaches [7-9] are essentially still designed for specific domains. AIGI-Holmes[10] (ICCV2025) pointed out in the Related Work section that existing MLLM methods are still *'domain-specific'*.
>
> Besides, the actual effectiveness of MLLMs highly depends on the visual modality. In our experiments, we found that although multimodal models incorporate language information, the performance improvements in classification and localization tasks almost entirely rely on the visual branch, with the language modality failing to effectively enhance discriminative capability.
>
> **Q3: Concerns of preprocessing.**
>
> A3: Yes, preprocessing is crucial for models when integrating models from different domains into a unified framework. Therefore, we deliberately decoupled the *Transform* component in the design of ForensicHub to accommodate diverse preprocessing requirements across different models and tasks (as mentioned in line 104 of the paper). ForensicHub inherently supports customizable preprocessing.
>
> However, tailoring the best augmentation strategy set for each model is very time-consuming. Therefore, we only preserve model-specific preprocessing where necessary. For example, models like CAT-Net, DTD, and FFDN retain their use of JPEG compression related features such as DCT coefficients and quantization tables (qtables).
>
> **As mentioned above, ForensicHub itself is designed with flexibility in preprocessing.** Therefore, in future updates, we will incorporate the optimal preprocessing strategies for each model to better unlock their full potential.
>
> **Q4: Deepfake and AIGC recent SoTAs need to be evaluated.**
>
> A4: **[Supplemental Experiments.]** We have added experiments on the deepfake DF40 dataset (NIPS24) and AIGC Chameleon dataset (ICLR 2025), along with evaluations of two recent Deepfake SoTAs: Sia (ECCV22) and Effort (ICML25), and two recent AIGC SoTAs: FatFormer (CVPR24) and CO-SPY (CVPR25). Results are shown below.
>
> |Model|DF40_CollabDiff|DF40_deepfacelab|DF40_heygen|Chameleon|
> |:-:|:-:|:-:|:-:|:-:|
> |ConvNeXt|0.935|0.795|0.744|0.626|
> |Capsule-Net|0.984|0.638|0.714|0.676|
> |MVSS-Net|0.983|0.699|0.456|0.725|
> |UnivFD|0.629|0.752|0.946|0.798|
> |DTD|0.494|0.477|0.517|0.631|
> |Sia|0.808|0.764|0.641|0.569|
> |Effort|0.995|0.894|0.949|0.604|
> |FatFormer|0.983|0.716|0.611|0.707|
> |CO-SPY|0.890|0.792|0.656|0.767|
>
> The upper table shows the AUC results of the five models (ConvNeXt (backbone), Capsule-Net (deepfake), MVSS-Net (imdl), DTD (document)) and four aforementioned‌ recent SoTAs above on the DF40 subset and the Chameleon dataset.
>
> |Model|FF-c40|CDFv2|DFD|Columbia|IMD2020|Autosplice|DF|GenImage|T-SROIE|OSTF|RTM|Avg.|
> |:-:|:-:|:-:|:-:|:-:|:-:|:-:|:-:|:-:|:-:|:-:|:-:|:-:|
> |Sia|0.629|0.584|0.671|0.653|0.483|0.626|0.593|0.748|0.610|0.677|0.574|0.622|
> |Effort|0.805|0.846|0.930|0.979|0.861|0.943|0.930|0.992|0.960|0.834|0.732|0.892|
> |FatFormer|0.842|0.770|0.866|0.199|0.585|0.784|0.941|0.999|0.983|0.806|0.751|0.758|
> |CO-SPY|0.819|0.780|0.875|0.460|0.716|0.779|0.940|0.989|0.969|0.836|0.748|0.829|
>
> The upper table shows four aforementioned‌ recent SoTAs AUC performance on test datasets across four domain. The strong performance of recent clip-based models like Effort and CO-SPY demonstrates the potential of clip-level analysis, offering valuable insights to researchers. We have tested all the test datasets, but due to space limitations (over 30+ test datasets), we only present a subset of the results. The complete data will be included in the final version if we have the chance.
>
> **[Maintenance Plan.]** *Considering the fast development of models and datasets of FIDL, we actually have a long-term maintenance plan for ForensicHub to include as many models and datasets as possible.* The plan can be briefed as:
>
> - We will add evaluations for baselines that have not yet been covered in DeepfakeBench and IMDLBenCo, and continuously follow updates from both benchmarks.
> - We will incorporate more SoTAs in the AIGC and Document domains, while actively tracking and integrating the latest datasets.
> - We aim to design more comprehensive evaluation metrics and establish a unified rank leaderboard under consistent scenarios for models across all domains, contributing to the overall advancement of the FIDL field.
>
> **Q5: Typos.**
>
> A5: Thank you for your feedback. We will correct these typos in the final version.
>
> If you have further issues, we are glad to discuss them with you sincerely.
>
> **References:**
>
> [1] Li, Y., & Lyu, S. (2018). Exposing deepfake videos by detecting face warping artifacts. arXiv preprint arXiv:1811.00656.
>
> [2] Matern, F., Riess, C., & Stamminger, M. (2019, January). Exploiting visual artifacts to expose deepfakes and face manipulations. In 2019 IEEE Winter Applications of Computer Vision Workshops (WACVW) (pp. 83-92). IEEE.
>
> [3] Zhou, P., Han, X., Morariu, V. I., & Davis, L. S. (2018). Learning rich features for image manipulation detection. In Proceedings of the IEEE conference on computer vision and pattern recognition (pp. 1053-1061).
>
> [4] Kwon, M. J., Nam, S. H., Yu, I. J., Lee, H. K., & Kim, C. (2022). Learning jpeg compression artifacts for image manipulation detection and localization. International Journal of Computer Vision, 130(8), 1875-1895.
>
> [5] Wang, Z., Bao, J., Zhou, W., Wang, W., Hu, H., Chen, H., & Li, H. (2023). Dire for diffusion-generated image detection. In Proceedings of the IEEE/CVF International Conference on Computer Vision (pp. 22445-22455).
>
> [6] Qu, C., Liu, C., Liu, Y., Chen, X., Peng, D., Guo, F., & Jin, L. (2023). Towards robust tampered text detection in document image: New dataset and new solution. In Proceedings of the IEEE/CVF Conference on Computer Vision and Pattern Recognition (pp. 5937-5946).
>
> [7] Xu, Z., Zhang, X., Li, R., Tang, Z., Huang, Q., & Zhang, J. FakeShield: Explainable Image Forgery Detection and Localization via Multi-modal Large Language Models. In The Thirteenth International Conference on Learning Representations.
>
> [8] Qu, C., Liu, J., Chen, H., Yu, B., Liu, J., Wang, W., & Jin, L. (2024). TextSleuth: Towards Explainable Tampered Text Detection. arXiv preprint arXiv:2412.14816.
>
> [9] Guo, X., Song, X., Zhang, Y., Liu, X., & Liu, X. (2025). Rethinking Vision-Language Model in Face Forensics: Multi-Modal Interpretable Forged Face Detector. In Proceedings of the Computer Vision and Pattern Recognition Conference (pp. 105-116).
>
> [10] Zhou, Z., Luo, Y., Wu, Y., Sun, K., Ji, J., Yan, K., ... & Ji, R. (2025). AIGI-Holmes: Towards Explainable and Generalizable AI-Generated Image Detection via Multimodal Large Language Models. arXiv preprint arXiv:2507.02664.

---

> ### Author Response · Authors · 2025-08-04
>
> Thank you for your thoughtful review. If additional experiments are needed, we’re happy to run them, though please note that our dataset covers four domains, so each experiment takes time, and if a request is raised too close to the deadline, we may not be able to complete it before the discussion window closes.

---

### Official Review · Reviewer_vowj · 2025-06-24

**Rating:** 6
**Confidence:** 4

**Summary:**

- This paper introduces ForensicHub, a unified benchmark and codebase for fake image detection and localization across four existing reaserch domains: Deepfake, IMDL, AIGC, and Document. It proposes a modular architecture that allows flexible integration of datasets, models, and evaluation tools. The benchmark includes 23 datasets, 42 models, 6 backbones, and 11 evaluation metrics.
- Further, it also introduces two new benchmarks for AIGC and Document tasks while being compatible with existing benchmark (DeepfakeBench and IMDLBenCo), and a unified IFF-Protocol is proposed for cross-domain fake image training/evaluation.
- The work offers a standardized platform to support reproducible and generalizable research in FIDL, and presents several meaningful insights into model design, dataset characteristics, and evaluation strategies.

**Additional Feedback:**

Given the potential impact of ForensicHub on multiple FIDL sub-domains, I am particularly interested in the authors’ plans for ongoing maintenance and updates to the benchmark.

**Dataset Code Accessibility:**

Yes

**Dataset Code Comments:**

The authors released the code with clear documentation, which supports reproducibility.

**Ethical Considerations:**

No, there are no or only very minor ethics concerns

**Final Justification:**

After reading through the authors rebuttal and other reviewers comments, I believe this is a solid work for the DB track.

This FIDL benchmark makes a great leap forward in addressing the critical unified fake image detection problem by establishing an arena to test the models behaviour across faking domains. I appreciate that the authors fully reproduced many close-sourced models and even constructed benchmarks for the Doc and AIGC domains from scratch. Their deduced insights align with my own experience and resolved my long-standing illusion about the efficacy of low-feature extractors, which are notable merits for a DB work.

The rebuttal adequately addressed my concerns through extensive experiments. With these experiments, this work contains largely complete SOTA models and datasets. Given the rapid advancements in the four FIDL subdomains, I also hope that the authors can keep updating the latest models and datasets in the future.

Based on the above, I raise my score to 6 and suggest accepting this work.

**Limitations Weaknesses:**

While ForensicHub is a valuable and ambitious effort, there are some areas for improvement.
- Although the authors implement several baselines for AIGC detection, the Section 4.1 may not be sufficiently comprehensive given the rapid development of generative models. Including more recent or diverse baselines would better support the paper’s goal of unification.
- In the IFF-Protocol evaluation (Section 5, Table 7), the paper primarily reports AUC scores. Providing additional metrics such as F1, TPR, or MCC could offer more nuanced insights and make the benchmark more useful for researchers across different domains. Addressing these points would further strengthen ForensicHub’s utility and completeness.

**Strengths Contributions:**

- This paper makes a valuable contribution to the FIDL field by addressing the long-standing issue of fragmentation across four sub-domains. The proposed ForensicHub unifies these tasks through a modular design that supports flexible integration of datasets, models, and evaluation tools. It covers a wide range of datasets and baselines, integrates two existing benchmarks (DeepfakeBench and IMDLBenCo), and introduces two new ones for AIGC and Document tasks, filling important gaps. Most statements are supported with comprehensive experiments.

- The IFF-Protocol enables cross-domain training and reveals valuable insights into model generalization, feature extractor effectiveness, and backbone performance.
- Overall, the paper is well-written, well-organized, and presents its contributions clearly to the fake image detect reaserch sociaty.

---

> ### Author Rebuttal · Authors · 2025-07-29
>
> Thank you for your valuable feedback and thoughtful comments. We sincerely appreciate your recognition of ForensicHub's **valuable contribution**, as well as your **endorsement of our IFF-Protocol and paper structure**. Here is our response to address your concerns.
>
> **Q1: Need more baseline for AIGC detection.**
>
> A1: **(Supplemental Experiments.)** As suggested, we have added one recent dataset in AIGC Detection, Chameleon (ICLR 2025), along with 3 recent SoTA methods: FatFormer (CVPR 2024), Effort (ICML25) and CO-SPY (CVPR 2025). Results are shown in the 2 tables below.
>
> |Model|DF40_CollabDiff|DF40_deepfacelab|DF40_heygen|Chameleon|
> |:-:|:-:|:-:|:-:|:-:|
> |ConvNeXt|0.935|0.795|0.744|0.626|
> |Capsule-Net|0.984|0.638|0.714|0.676|
> |MVSS-Net|0.983|0.699|0.456|0.725|
> |UnivFD|0.629|0.752|0.946|0.798|
> |DTD|0.494|0.477|0.517|0.631|
> |FatFormer|0.983|0.716|0.611|0.707|
> |Effort|0.995|0.894|0.949|0.604|
> |CO-SPY|0.890|0.792|0.656|0.767|
>
> The upper table shows the AUC results of the five models (ConvNeXt (backbone), Capsule-Net (deepfake), MVSS-Net (imdl), DTD (document)) and 2 AIGC recent SoTAs on the Chameleon dataset (the last column). It can be observed that Chameleon is more challenging compared to GenImage and DiffusionForensics. Moreover, AIGC-domain models UnivFD and CO-SPY achieve strong performance.
>
> |Model|FF-c40|CDFv2|DFD|Columbia|IMD2020|Autosplice|DF|GenImage|T-SROIE|OSTF|RTM|Avg.|
> |:-:|:-:|:-:|:-:|:-:|:-:|:-:|:-:|:-:|:-:|:-:|:-:|:-:|
> |FatFormer|0.842|0.770|0.866|0.199|0.585|0.784|0.941|0.999|0.983|0.806|0.751|0.758|
> |Effort|0.805|0.846|0.930|0.979|0.861|0.943|0.930|0.992|0.960|0.834|0.732|0.892|
> |CO-SPY|0.819|0.780|0.875|0.460|0.716|0.779|0.940|0.989|0.969|0.836|0.748|0.829|
>
> The upper table shows 3 aforementioned‌ recent SoTAs AUC performance on test datasets across four domain. The superior performance of the Effort model in the IMDL can provide valuable insights for researchers. We have tested all the test datasets, but due to space limitations (over 30+ test datasets), we only present a subset of the results. The complete data will be included in the final version if we have the chance.
>
> **(Maintenance Plan.)** *Considering the fast development of models and datasets of FIDL, we actually have a long-term maintenance plan for ForensicHub to include as many models and datasets as possible.* The plan can be briefed as:
>
> - We will add evaluations for baselines that have not yet been covered in DeepfakeBench and IMDLBenCo, and continuously follow updates from both benchmarks.
> - We will incorporate more SoTAs in the AIGC and Document domains, while actively tracking and integrating the latest datasets.
> - We aim to design more comprehensive evaluation metrics and establish a unified rank leaderboard under consistent scenarios for models across all domains, contributing to the overall advancement of the FIDL field.
>
> Following the above plan, ForensicHub will eventually incorporate all models and datasets across the four domains, which is only a matter of time.
>
> **Q2: Additional metrics for IFF-Protocol.**
>
> A2: We will include the ranking leaderboards for all models, aiming to provide researchers with insights into model architectures and fair comparisons. In accordance with the rebuttal policy (images and links are forbidden), we will release the rank leaderboard in the final version. **The leaderboards present seven evaluation metrics across all test datasets: F1, AUC, AP, MCC, TNR, TPR, and Accuracy.** We will continue to maintain these leaderboards and regularly update them with the latest state-of-the-art models from each domain.
>
> Since the ranking table includes all test datasets (over 30 in total), below is the abbreviated version of the ranking table：
>
> | Rank | Model         | Avg F1 | FF++c40 | CelebDF | ... |
> |------|---------------|--------|---------|---------|-----|
> | 1    | Swin          | 0.7875 | 0.7423  | 0.8076  | ... |
> | 2    | FFDN          | 0.7823 | 0.6972  | 0.7994  | ... |
> | 3    | ConvNeXt      | 0.7752 | 0.8022  | 0.7588  | ... |
> | 4    | Segformer     | 0.7624 | 0.8786  | 0.7364  | ... |
> | ...  | ...           | ...    | ...     | ...     | ... |
>
> If you have further issues, we are glad to discuss them with you sincerely.

---

> > ### Comment · Reviewer_vowj · 2025-08-05
> >
> > As all of my questions have been addressed. Thanks. I will rasie my score.

---

> > > ### Author Response · Authors · 2025-08-06
> > >
> > > Thank you for your positive feedback and for taking the time to review our work. We appreciate your recognition. We will make every effort to maintain this project on GitHub and contribute to the research community.

---

### Official Review · Reviewer_JrML · 2025-07-03

**Rating:** 4
**Confidence:** 5

**Summary:**

This paper introduces ForensicHub, a unified benchmark and codebase for universal fake image detection and localization, covering four classic tasks in media forensics: deepfake detection, image manipulation detection/localization, AI-generated image detection, and document image manipulation localization. This work builds a universal benchmark and codebase that can be used for different protocols in detection.

**Dataset Code Accessibility:**

Partly

**Ethical Considerations:**

No, there are no or only very minor ethics concerns

**Final Justification:**

The authors have addressed some of my concerns, I decide to maintain my original score.

**Limitations Weaknesses:**

1. Many of the datasets and methods included in the benchmark are outdated. For example, Table 2 primarily references deepfake datasets from 5–7 years ago. Recent and more challenging datasets—such as DF40 (NeurIPS 2024) and AIFace (CVPR 2025)—are notably absent. Similarly, the benchmark omits several recent SOTA detectors. While UnivFD is included for AIGC detection, newer and more advanced methods like Effort (ICML 2025) and CLIP-C2P (AAAI 2025) should be considered.
2. While the paper presents evaluations under different protocols, it lacks in-depth analysis or insightful observations. The work feels more like an engineering artifact than a scientific study. Deeper discussion and interpretation of results are necessary to elevate the contribution.
3. The number of AIGC datasets is both small and outdated. High-quality recent datasets, such as Chameleon (ICLR 2025), should be included to better reflect current challenges in AIGC detection.
4. The IMDL task is only used for training, without a corresponding test set. This unbalanced design is questionable. The rationale behind this setup should be clarified or revised.
5. The paper does not explore whether detectors designed for blending-based deepfakes (e.g., SBI, which detects blending boundaries) could be effective for IMDL-like tasks. Since blending artifacts are a known forensic clue, evaluating their transferability would be valuable.
6. Vision-language models (VLMs) like CLIP, which have shown promise in forensic detection, are not considered. Strong recent models like Effort (ICML 2025) and FatFormer (CVPR 2024) should be included, as many studies emphasize their importance in generalizable detection.
7. The paper does not discuss which model architectures are best suited for cross-task forensic learning. A comparative analysis of architectural choices would be insightful.
8. No visualizations (e.g., t-SNE plots, heatmaps) are provided to understand the feature space or model behavior across tasks. These would help illustrate task similarities or divergences.
9. The paper does not examine whether training on all forensic tasks simultaneously yields better generalization, or whether conflicting patterns between tasks hinder performance. Prior work on multi-task learning (e.g., UCF, ICCV 2023) could offer insight. It's important to investigate whether common features exist across tasks and how to address potential task conflicts.

**Strengths Contributions:**

1. This is the first work, in my knowledge, to unify different yet similar forensic tasks in one benchmark/codebase.
2. The unified framework allows the community to examine whether detectors trained on domain-specific tasks—such as deepfake detection—can generalize to other types of forensic tasks. This offers potential insights into transferability.
3. The code is modular and extendable, making it easier for researchers to implement and test new detection models across multiple tasks within a single infrastructure.

---

> ### Author Rebuttal · Authors · 2025-07-29
>
> Thank you for your valuable feedback and thoughtful comments. We sincerely appreciate your recognition of ForensicHub as **the first unified image forensic benchmark, its potential to provide transferability insights, and the modularity and extensibility of our code**. Here is our response to address your concerns.
>
> **Q1: Need more recent dataset and model (point 1,3,6).**
>
> A1: **[Supplemental Experiments.]** For point 1, 3 and 6, thank you for your valuable suggestion. As suggested, we have added experiments on the DF40 and Chameleon dataset, along with evaluations of two recent Deepfake SoTAs: Sia (ECCV22) and Effort (ICML25), and two recent AIGC SoTAs: FatFormer (CVPR24) and CO-SPY (CVPR25). Experiments are shown in the 2 tables below:
>
> |Model|DF40_CollabDiff|DF40_deepfacelab|DF40_heygen|Chameleon|
> |:-:|:-:|:-:|:-:|:-:|
> |ConvNeXt|0.935|0.795|0.744|0.626|
> |Capsule-Net|0.984|0.638|0.714|0.676|
> |MVSS-Net|0.983|0.699|0.456|0.725|
> |UnivFD|0.629|0.752|0.946|0.798|
> |DTD|0.494|0.477|0.517|0.631|
> |Sia|0.808|0.764|0.641|0.569|
> |Effort|0.995|0.894|0.949|0.604|
> |FatFormer|0.983|0.716|0.611|0.707|
> |CO-SPY|0.890|0.792|0.656|0.767|
>
> The upper table shows AUC of the five models (one backbone and four domain-SoTA) and four aforementioned‌ recent SoTAs on the DF40 subset and the Chameleon dataset. Combined with Table 7 in the paper (AUC of all models), it can be observed that ConvNeXt still demonstrates superior potential. Among AIGC detection datasets, Chameleon is more challenging compared to GenImage and DiffusionForensics.
>
> |Model|FF-c40|CDFv2|DFD|Columbia|IMD2020|Autosplice|DF|GenImage|T-SROIE|OSTF|RTM|Avg.|
> |:-:|:-:|:-:|:-:|:-:|:-:|:-:|:-:|:-:|:-:|:-:|:-:|:-:|
> |Sia|0.629|0.584|0.671|0.653|0.483|0.626|0.593|0.748|0.610|0.677|0.574|0.622|
> |Effort|0.805|0.846|0.930|0.979|0.861|0.943|0.930|0.992|0.960|0.834|0.732|0.892|
> |FatFormer|0.842|0.770|0.866|0.199|0.585|0.784|0.941|0.999|0.983|0.806|0.751|0.758|
> |CO-SPY|0.819|0.780|0.875|0.460|0.716|0.779|0.940|0.989|0.969|0.836|0.748|0.829|
>
> The upper table shows four aforementioned‌ recent SoTAs AUC performance on test datasets across four domains. We have tested all the test datasets, but due to space limitations (over 30+ test datasets), we only present a subset of the results. The complete data will be included in the final version if we have the chance.
>
> **[AIFace and CLIP-C2P Not Included.]** Experiments on the AIFace dataset (the size exceeds 200 GB) are not included due to the rebuttal time constraint. Although we tried various acceleration methods, it would take over 6 days to download on our server. Likewise, CLIP-C2P lacks training code and preprocessing the training data to obtain captions for CLIP-C2P is extremely time-consuming under IFF-Protocol of ForensicHub. We will include them in future updates and maintenance of ForensicHub.
>
> For short, we have conducted extra experiments on DF-40 and Chameleon datasets, and implemented 2 recent Deepfake SoTAs and 2 recent AIGC SoTAs.
>
> **[Maintenance Plan.]** *Considering the fast development of models and datasets of FIDL, we actually have a long-term maintenance plan for ForensicHub to include as many models and datasets as possible.*
>
> **Q2: Paper lacks in-depth analysis or insightful observations (point 2).**
>
> A2: We elaborately designed additional experiments to **analyse‌ two widely-concerned but less-explored problems**, and **deduced counter-intuitive insights and prescient observations** in Section 4,5,6 of the paper. In short, our in-depth analysis and observations are:
>
> - Based on ForensicHub, we explored two key issues: low-level feature extractors and cross-domain model transference. These two topics are widely concerned but severely underexplored. We conducted a series of experiments and provided analysis and answers to both.
> - Building on the above experiments, we further conducted analyses that deduced 3 counter-intuitive insights. For example, shallow-fusion features showed little to no benefit, and simple backbone designs like ConvNeXt can outperform more complex SoTAs. These counter-intuitive insights challenge the consensuses made in many previous studies.
> - Observing our experiments and insights, we concluded several potential directions for future improvement. For example, the generalization issues in current evaluation protocols for AIGC and Document, and dataset complexity challenges. These observations validate and align with current active research efforts in these areas.
>
> We will emphasize our insights more clearly in the final version.
>
> **Q3: Concerns of IMDL train and test datasets (point 4).**
>
> A3: We would like to address your concern regarding the IMDL datasets in two parts:
>
> - **We did include IMDL datasets for testing.** As shown in Table 7 of the paper, we used Columbia, IMD2020, and AutoSplice for testing, which are all IMDL datasets.
> - **A common practice of the split protocol in IMDL is using a larger dataset (like CASIAv2) as the only train set, and other datasets as test sets.** IMDLBenCo (NIPS24) indicates that this common practice is due to the relatively small size of mainstream IMDL datasets, which makes models prone to overfitting. We followed such common practice in our experiments.
>
> **Q4: Effectiveness‌ of blending-based deepfakes detectors for IMDL-like tasks (point 5).**
>
> A4: You are correct that blending-based techniques like SBI are helping in IMDL. In fact, the IMDL field has already widely adopted more general blending artifact enhancement techniques, **which have already been proven to be effective for artifacts capture by previous works (IMDLBenCo), so we didn't test the Deepfake blending-based techniques on IMDL models in ForensicHub**.
>
> IMDLBenCo (NIPS24) integrates methods like RandomCopyMove and RandomInpainting. These methods generate controllable manipulated regions randomly in an image, producing diverse artifacts. IMDLBenCo proved that such methods improve models performance.
>
> **Q5: Best model architectures for cross-task forensic learning (point 7).**
>
> A5: **In FIDL, a model's 'model architecture' typically consists of two components: the backbone and the low-level feature extractor.** Therefore, we will address your concern from these two perspectives. If our interpretation of 'model architecture' is incorrect, please let us know, and we will further respond to address your concerns.
>
> - **From a perspective of backbone**, we discussed in *line 220* of the paper that ConvNeXt and Swin Transformer demonstrate strong performance under cross-domain settings, even surpassing most domain-specific SoTAs. CNN-based ConvNeXt and Transformer-based Swin Transformer perform almost equally well, indicating that both architectures are currently suitable for FIDL.
>
> - **From a perspective of feature extractor**, in Section 6 of the paper, we conducted experiments and found that under cross-domain settings, almost all shallow layer concatenation-based low-level feature extractors fail to effectively capture artifacts. This provides a counter-intuitive yet valuable insight for the FIDL community.
>
> **Q6: Visualizations (point 8).**
>
> A6: We use Grad-CAM to visualize the heatmaps of models from the four domains (Capsule-Net (deepfake), MVSS-Net (imdl), UnivFD (aigc),  DTD (document)) on datasets from each domain, aiming to explore their attention regions. In compliance with the rebuttal policy, we can't provide images, but we will present the visualizations in the final version.
>
> **The results show that models from different domains exhibit both similarities and differences** in their attention regions for the same image. For example, on Deepfake datasets, Capsule-Net focuses more on specific eyes and nose facial features, while MVSS-Net and DTD focus on larger facial regions. On Document datasets, Capsule-Net, MVSS-Net, and UnivFD tend to focus on the overall tampered text regions, whereas DTD pays more attention to subtle traces of manipulation, such as the edges and curves of characters.
>
> **Q7: Common features and conflicting patterns across domains (point 9).**
>
> A7: We selected two IMDL models: MVSS-Net and IML-ViT, and **used IFF-Protocol weights (where models are trained across-domain) as pretrained weights**. These models were then trained on the IMDL task to investigate whether the artifacts learned across domains could benefit finetuning within a single domain. Results are shown in the table below.
>
> |Method|Coverage|Columbia|NIST16|CASIAv1|IMD2020|Avg.|
> |:-:|:-:|:-:|:-:|:-:|:-:|:-:|
> |MVSS-Net (Image-Net)|0.259|0.386|0.246|0.534|0.279|0.341|
> |MVSS-Net (IFF-Protocol)|0.268|0.395|0.259|0.562|0.292|0.355|
> |IML-ViT (Image-Net)|0.435|0.780|0.331|0.721|0.327|0.519|
> |IML-ViT (IFF-Protocol)|0.427|0.767|0.279|0.715|0.351|0.508|
>
> **The results show that there are both common features and conflicting patterns across domains.** Using IFF-Protocol weights, MVSS-Net achieved improvements across all IMDL datasets, while IML-ViT improved on IMD2020. This indicates that there are indeed similar artifacts across domains, and certain models could leverage such common features to achieve better generalization. Meanwhile, the performance drop of IML-ViT on four datasets suggests that it may have learned conflicting patterns from other domains under IFF-Protocol, which are misaligned with the IMDL task.
>
> We hypothesize that the difference stems from the **presence or absence of low-level feature extractors**. Models with dedicated feature extractors (BayarConv in MVSS-Net) may learn relatively common features across domains, while those without (IML-ViT) may instead capture conflicting patterns. We will further explore these common features and conflicting patterns across domains in future maintenance and updates of ForensicHub.
>
> If you have further issues, we are glad to discuss them with you sincerely.

---

### Author Response · Authors · 2025-08-02
**General Response**

**General Response:**

We sincerely appreciate the reviewers’ insightful comments and constructive feedback on our manuscript. We are pleased to receive positive recognition from most of the reviewers. In particular, we are delighted that the reviewers found our benchmark “unifying different forensic tasks into a single benchmark/codebase” (reviewer JrML & reviewer vowj), “valuable insights into cross-domain generalization and transferability” (reviewer JrML & reviewer vowj), and “modular and extensible framework design supporting flexible integration of datasets, models, and evaluators” (all reviewers).

Based on the reviews, we provide below a general response to the commonly raised concerns, followed by individual responses to address each reviewer’s specific questions.

(1) **Regarding the experimental results, we have taken the following actions:**
- For all reviewers, we conducted new experiments on the DF40 and Chameleon datasets, and included evaluations of four recent deepfake and AIGC SoTA models (Sia, Effort, FatFormer, and CO-SPY).
- For reviewer JrML, we added cross-domain transferability analysis under the IFF-Protocol, providing both quantitative results and insightful interpretations of model behavior.
- For reviewer JrML, we also conducted visualization analysis (e.g., Grad-CAM) to investigate models’ attention regions


(2) **In response to questions about the motivation, benchmark design, and technical details, we have made the following clarifications:**
- For Reviewer JrML, we clarified the model architectural choices for cross-domain forensic learning, and the reasoning for IMDL training protocol design.
- For Reviewer vowj, we elaborated on the benchmark coverage, maintenance plan, and explained the the leaderboard design for model comparison.
- For Reviewer 4Hjf, we clarified the core motivation behind unifying these forensic domains based on shared artifact detection goals, and addressed misunderstandings about the relevance of MLLMs. We also explained how preprocessing diversity is handled through ForensicHub’s modular pipeline.

(3) **Regarding the valuable suggestions on presentation and organization from all reviewers, we will incorporate them and make the necessary revisions in the final version of the paper.**

Once again, we thank you for your thoughtful and constructive suggestions -- they have greatly helped us improve the quality and clarity of the paper. During the permitted discussion time, we would be glad to post any extended replies on the forum. Please don't hesitate to let us know if you would like any further clarifications or if there are minor points we can address. We'd be happy to provide additional information to help convey the merits of our work. We truly appreciate your feedback.

Yours sincerely,
Authors of #663

---

### Author Response · Authors · 2025-08-09
**General Response to Reviewers: Gratitude for Your Feedback & Inquiry on Further Questions (Deadline Approaching)**

Dear Reviewers,​

Thank you all for your insightful responses to our rebuttal. We are pleased to know that we have addressed all questions for reviewer vowj. **We are particularly delighted that reviewer vowj affirmed our rebuttal and considered a score increase, as well as reviewer JrML and vowj giving us highly positive feedback right from the start.** All your positive feedback not only acknowledges our effort during rebuttal time, but also greatly helps us improve the manuscript.

**As the discussion DDL is approaching, please let us know if there are any further questions or concerns**—we will promptly provide additional clarifications. Again, thank you for your recognition and endeavor in improving our paper.​

Best regards,​

Authors of submission #663

---

### Decision · Program_Chairs · 2025-09-18

**Decision:**

Accept (poster)

**Comment:**

This paper introduces ForensicHub, a comprehensive benchmark and codebase aiming to unify research across four major subdomains of fake image detection: deepfake detection, image manipulation detection/localization (IMDL), AI-generated image (AIGC) detection, and document image manipulation localization. The work provides a modular, extensible architecture, integrates numerous datasets and models, and proposes the IFF-Protocol for cross-domain training/evaluation. The code and datasets are made publicly available, supporting reproducibility. All reviewers argree that ForensicHub is a technically solid and impactful resource for the community.